# *Tbx1* haploinsufficiency leads to local skull deformity, paraflocculus and flocculus dysplasia, and motor-learning deficit in 22q11.2 deletion syndrome

Tae-Yeon Eom [1], J. Eric Schmitt [2,3], Yiran Li [1], Christopher M. Davenport[1], Jeffrey Steinberg [4], Audrey Bonnan[5], Shahinur Alam[1,6], Young Sang Ryu[4], Leena Paul[1], Baranda S. Hansen [7], Khaled Khairy [1,6], Stephane Pelletier [8], Shondra M. Pruett-Miller [7,9], David R. Roalf [3], Raquel E. Gur[3], Beverly S. Emanuel [10,11], Donna M. McDonald-McGinn[10,11,12], Jesse N. Smith[13], Cai Li [13], Jason M. Christie [5,14], Paul A. Northcott [1] & Stanislav S. Zakharenko [1]✉

Neurodevelopmental disorders are thought to arise from intrinsic brain abnormalities. Alternatively, they may arise from disrupted crosstalk among tissues. Here we show the local reduction of two vestibulo-cerebellar lobules, the paraflocculus and flocculus, in mouse models and humans with 22q11.2 deletion syndrome (22q11DS). In mice, this paraflocculus/flocculus dysplasia is associated with haploinsufficiency of the *Tbx1* gene. *Tbx1* haploinsufficiency also leads to impaired cerebellar synaptic plasticity and motor learning. However, neural cell compositions and neurogenesis are not altered in the dysplastic paraflocculus/flocculus. Interestingly, 22q11DS and *Tbx1*⁺ᐟ⁻ mice have malformations of the subarcuate fossa, a part of the petrous temporal bone, which encapsulates the paraflocculus/flocculus. Single-nuclei RNA sequencing reveals that *Tbx1* haploinsufficiency leads to precocious differentiation of chondrocytes to osteoblasts in the petrous temporal bone autonomous to paraflocculus/flocculus cell populations. These findings suggest a previously unrecognized pathogenic structure/function relation in 22q11DS in which local skeletal deformity and cerebellar dysplasia result in behavioral deficiencies.

The complex 22q11.2 deletion syndrome (22q11DS) is caused by hemizygous microdeletions of human chromosome 22 and occurs at a frequency of ~1 in every 2148 live births[1–4], making it the most common deletion in humans. It is a heterogeneous disorder associated with a wide spectrum of phenotypes that affects the early morphogenesis of many organs and systems, causing cardiac abnormalities, thymic hypoplasia, palatal defects, and structural central nervous system (CNS) anomalies,

among others[5–8]. In addition, patients with 22q11DS have learning disabilities; their cognitive abilities decline with age and are co-morbid with increased risk of psychiatric disease[6,9], including schizophrenia (SCZ), autism-spectrum disorders, attention deficit hyperactivity disorder, and Parkinson's disease[10–22]. SCZ arises in 25–30% of patients with 22q11DS during adolescence or early adulthood, making this microdeletion one of the strongest predictors of SCZ. Symptoms of 22q11DS-related SCZ

are indistinguishable from those of other types of SCZ[17,23,24], suggesting that 22q11DS affects the same biological mechanisms affected in the general SCZ population. The brain regions and neural circuits that cause the neuropsychiatric symptoms of 22q11DS are mostly unknown. Identifying culprit brain regions and circuits has been facilitated by studying mouse models of 22q11DS (22q11DS mice), which carry a heterozygous deletion of the region encompassing most of the disease-crucial genes[24–27]. Magnetic resonance imaging (MRI) has shown a whole-brain and regional volumetric reduction in patients with 22q11DS[28–31] and in 22q11DS mice[32,33]. However, most neuroanatomic alterations are widespread and subtle-to-modest in strength. More importantly, the 22q11DS genes, downstream mechanisms responsible for these volumetric alterations, and the functional consequences of such alterations are mostly unknown.

Here we show a local and substantial reduction (dysplasia) of the paraflocculus and flocculus (PF/F), bulb-like lobules in the ventrolateral part of the vestibulo-cerebellum[34], in 22q11DS mice. This PF/F dysplasia is accompanied by the abnormal development of the specific part of the petrous temporal bone (PTB) that houses the PF/F, the subarcuate fossa (SF). We also confirm a specific reduction of these cerebellar lobules in human subjects with this disease. This is consistent with increasing evidence indicating that the cerebellum is involved in cognition, multimodal sensory and sensory motor integration, and other functions that are affected in neuropsychiatric disease[35,36]. In this work, we have also identified the causal gene and suggested the mechanisms responsible for the localized cerebellar abnormality in 22q11DS.

## Results

### 22q11.2 microdeletion specifically reduces the PF/F in mice and humans

We examined brain structure at different ages, ranging from postnatal day (P) 7 to adulthood (8 months), in *Df(16)1/+* murine models of 22q11DS[25] by using MRI in vivo. One noticeable structural difference was that the PF/F volume was considerably smaller in the *Df(16)1/+* mice compared to wild-type (WT) littermates (Fig. 1a, b and Supplementary Movies 1, 2). We, therefore, focused this investigation on the cerebellum. The PF/F dysplasia was detectable throughout development, starting at P7 and continuing into adulthood (Fig. 1a–d), and did not depend on sex or laterality (Supplementary Fig. 1a, b). This dysplasia in *Df(16)1/+* mice was specific to the PF/F; neither the total-brain nor the cerebellar volume was smaller in the 22q11DS models compared to WT littermates (Supplementary Fig. 1c, d). Follow-up measurements of the PF/F volumes by high-resolution MRI ex vivo revealed that the PF/F volume was substantially smaller in *Df(16)1/+* mice compared to WT littermates. Manual measurement of the outer PF/F volumes revealed a 72% decrease (WT: 2.93 mm³ ± 0.15 mm³; *Df(16)1/+*: 0.83 mm³ ± 0.07 mm³, p < 0.0001) (Supplementary Fig. 1e, f), and automated machine learning–based measurement[37] also showed a 70% decrease (WT: 3.27 mm³ ± 0.18 mm³; *Df(16)1/+*: 0.97 mm³ ± 0.08 mm³, p < 0.0001) of the PF/F in 22q11DS models (Supplementary Fig. 1g, h). The specificity of the PF/F dysplasia in 22q11DS models was further confirmed when we compared different cerebellar regions between *Df(16)1/+* and WT mice. Although we observed a substantial reduction in the *PF/F*, no volumetric differences were detected in Crus I, Crus II, or Vermis IV/V regions between *Df(16)1/+* and age-matched WT mice (Fig. 1e).

To investigate whether the same PF/F dysplasia phenotype observed in mice also occurs in humans, we examined MRI data from 80 individuals with 22q11DS and from 68 typically developing (TD) individuals (Fig. 1f–k). Two independent approaches, manual measurements of the defined regions of interest (ROIs) (Fig. 1f, g) and voxel-based morphometry (VBM) (Fig. 1j, k, Supplementary Fig. 2a–c, and Table 1), revealed that the PF/F was disproportionally affected in

individuals with 22q11DS compared to TD controls. The PF/F dysplasia in those with 22q11DS did not depend on sex or laterality (Supplementary Fig. 2d, e). The mean PF/F volume was 28.6% smaller in subjects with 22q11DS relative to TD controls (Fig. 1g, TD: 745.4 mm³ ± 19.4 mm³; 22q11DS: 558.85 mm³ ± 15.06 mm³, p < 0.0001). Unlike our findings in mouse models, the total-brain and cerebellar volumes in human subjects with 22q11DS were smaller relative to TD controls (11.7 and 17.3% reduction, respectively) (Supplementary Fig. 2f, g). Unlike the mouse data, all human cerebellar lobules (including the flocculonodular lobe) and cerebellar white matter were smaller in 22q11DS (Supplementary Fig. 2h). However, the PF/F was affected more than other parts of the cerebellum. Although the PF/F and cerebellar volumes were moderately correlated (Fig. 1h, r = 0.60; 95% CI [0.49–0.69]; TD only: r = 0.50, 95% CI [0.29–0.66]; 22q11DS only: r = 0.31, 95% CI [0.09–0.50]), group differences in PF/F volumes remained statistically significant even after adjusting for total cerebellar volume (Fig. 1i). Moreover, the VBM approach showed that the largest significant clusters were observed at the bilateral pontocerebellar junctions near the PF/F (Fig. 1j, k and Supplementary Fig. 2a–c). These significant clusters also included the lateral gray matter of right lobules V and VI and bilateral Crus I. Much smaller significant clusters were observed symmetrically in the deep cerebellar white matter near the dentate nucleus. Thus, the PF/F was specifically reduced in mouse models and humans with 22q11DS but at a larger degree in mice than in humans. This interspecies difference could be due to disparate anatomy or development of the PF/F (see Discussion).

### *Tbx1* haploinsufficiency underlies the PF/F dysplasia in 22q11DS mice

We validated our finding of PF/F dysplasia in *Df(16)1/+* mouse models[25] in other 22q11DS mouse models. Mice carrying a microdeletion equivalent to the most common LCR22A-LCR22D 3.0-Mb microdeletion in humans (Del(3.0 Mb)/+ mice[38]) and mice carrying a microdeletion equivalent to the LCR22A-LCR22B 1.5-Mb microdeletion (*LgDel/+* mice[26]) also had a substantial decrease in the PF/F volume compared to their respective WT littermates (Fig. 2a, b). These data further confirmed the robustness of our finding of the PF/F reduction in 22q11DS models.

To identify the gene(s), the haploinsufficiency of which causes the PF/F dysplasia in 22q11DS mice, we took advantage of a panel of mutant mice carrying smaller subregional hemizygous deletions or hemizygous deletion of an individual gene within the *Df(16)1* genomic region (Fig. 2a)[39]. We focused on the *Df(16)1* region because the PF/F reduction in *LgDel/+* mice carrying a microdeletion of 28 genes and that of *Df(16)1/+* mice carrying a microdeletion of 23 genes were of similar degree. Moreover, the PF/F dysplasia in *Del(3.0 Mb)/+* mice, carrying a microdeletion of 39 genes, was not as severe as that of *LgDel/+* or *Df(16)1/+* mice (Fig. 2b). These data suggest that the causal gene(s) should map within the *Df(16)1* genomic region.

*Df(16)2/+* mice, *Df(16)3/+* mice, and *Df(16)4/+* mice but not *Df(16)5/+* mice or *Znf74l-Ctp/+* mice showed the PF/F dysplasia (Fig. 2b). These results narrowed the culprit genomic region. However, the PF/F dysplasia in *Df(16)2/+* mice was less severe (32.8% ± 22.4% reduction) than that in *Df(16)3/+* mice (56.4% ± 13.9% reduction, t₁₃.₄₈ = 2.58, p = 0.02) or *Df(16)4/+* mice [56.6% ± 18.4% reduction, t₁₀.₅₆ = 2.55, p = 0.03; calculated as PF/F reduction = (PF/F_{mutant} − mean PF/F_{WT})/mean PF/F_{WT}]. This suggested that at least two genes could be involved in the PF/F phenotype––a gene with a smaller effect located within the *Df(16)2/+* region and a gene with a larger effect located within the *T10-Sept5* region containing eight protein-encoding genes. To narrow the large-effect gene candidates, we first produced and validated a mutant mouse carrying a hemizygous deletion of three genes: *Arvcf, Comt*, and *Txnrd2* (*Arvcf-Txnrd2/+* mice) (Supplementary Fig. 3a). We also produced *T10+/−* mice (Supplementary Fig. 3b). We then measured the PF/F volume in *Arvcf-Txnrd2/+* mice; *T10+/−* mice; previously produced

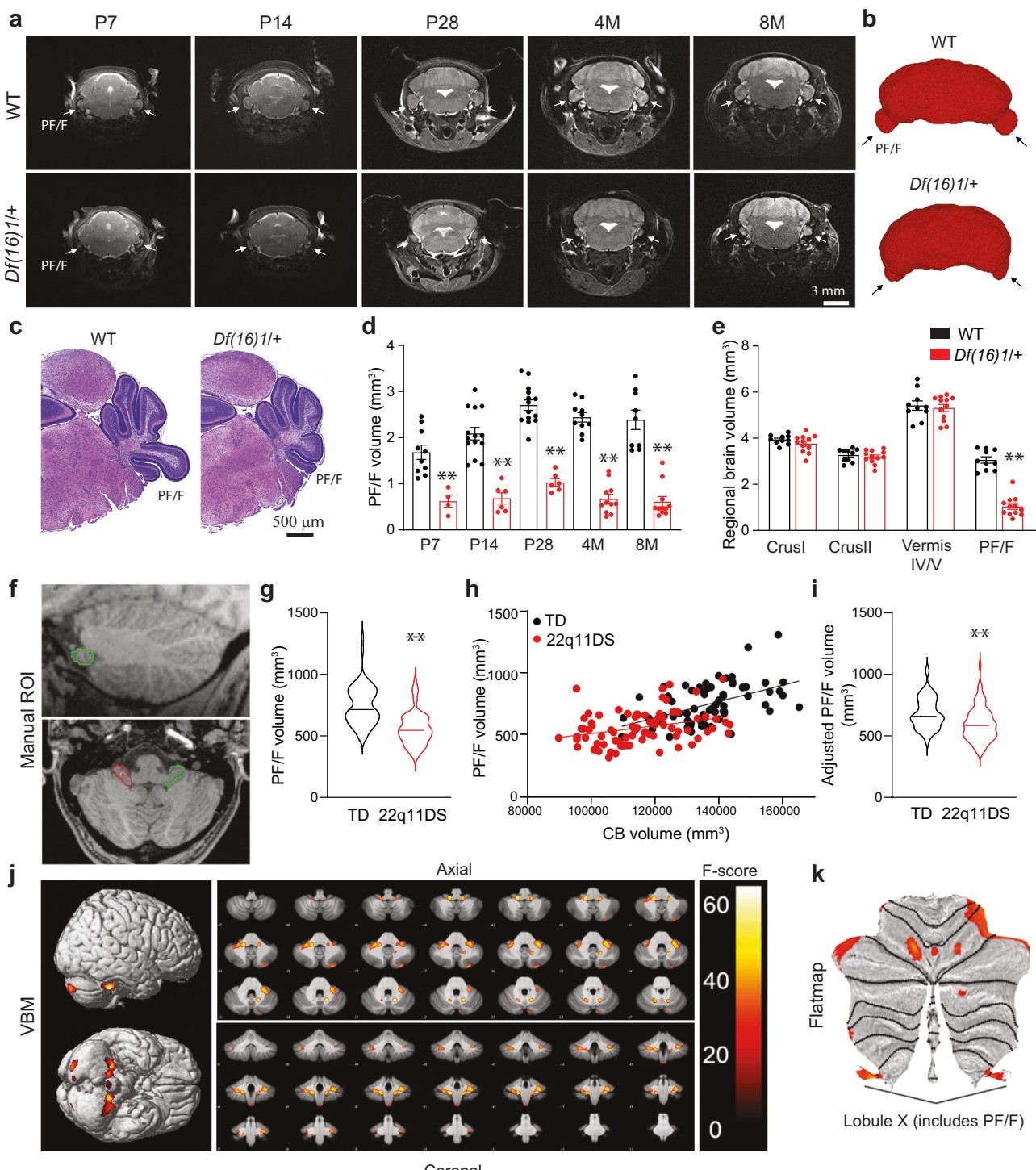

**Fig. 1 | Paraflocculus/flocculus dysplasia in mouse models and human subjects with 22q11DS. a** Representative magnetic resonance images (MRIs) of the cerebellum containing the paraflocculus/flocculus (PF/F, arrows) in wild-type (WT) and *Df(16)1/+* mice at different ages. **b** Representative 3D rendering of MRI data from 2-month-old WT and *Df(16)1/+* cerebella. Arrows, PF/F. **c** Representative images of hematoxylin and eosin staining of postnatal day (P) 7 WT and *Df(16)1/+* cerebella. **d** Average PF/F volumes measured by MRI in WT (black, P7: $n = 10$, P14: $n = 14$, P28: $n = 14$, 4-month [M]: $n = 10$, 8 M: $n = 9$) and *Df(16)1/+* (red, P7: $n = 4$, P14: $n = 6$, P28: $n = 6$, 4 M: $n = 11$, 8 M: $n = 11$) mice. Two-way ANOVA (Holm–Sidak's post hoc): $F_{4,85} = 5.47$, **$p = 0.0006$. **e** Average regional volumes of the cerebellum manually measured by MRI in 2-month-old WT ($n = 3$) and *Df(16)1/+* ($n = 4$) mice. Two-way ANOVA (Holm–Sidak's post hoc): $F_{3,20} = 7.76$: **$p = 0.0013$ (Crus I: $p = 0.95$; Crus II: $p = 0.99$; Vermis IV/V: $p = 0.99$; PF/F: $t_5 = 5.92$, **$p = 0.002$). **f** Representative MRIs of

a human cerebellum with manual tracings of the regions of interest (ROIs, red, left side; green, right side), i.e., the PF/F in sagittal (top) and axial (bottom) views. **g** PF/F volume in human subjects with 22q11DS (n = 80) and in typically developing (TD; $n = 68$) controls. Two-tailed Student's $t$-test; $t_{146} = 7.72$, **$p < 0.0001$, 95% CI = −234-138.6. **h** PF/F volume as a function of the cerebellar (CB) volume in TD ($n = 68$) and 22q11DS ($n = 80$); TD: Pearson's $r = 0.50$, $p < 0.0001$, 22q11DS: Pearson's $r = 0.31$, $p = 0.01$. **i** PF/F volume after adjusting for CB volume in each group. Wald test, $t_{145} = -3.268$, **$p = 0.001$. **j, k** Probability maps from cerebellar voxel-based morphometry (VBM), projected onto both whole-brain renders (**j**, left), selected axial and coronal slices (**j**, right), and a cerebellar flatmap (**k**). Note that SUIT VBM excludes the cerebrum from the analysis. Data are presented as the mean ± SEM in (**d**, **e**) and as violin plots with median values in (**g**, **i**). Source data are provided as a Source Data file.

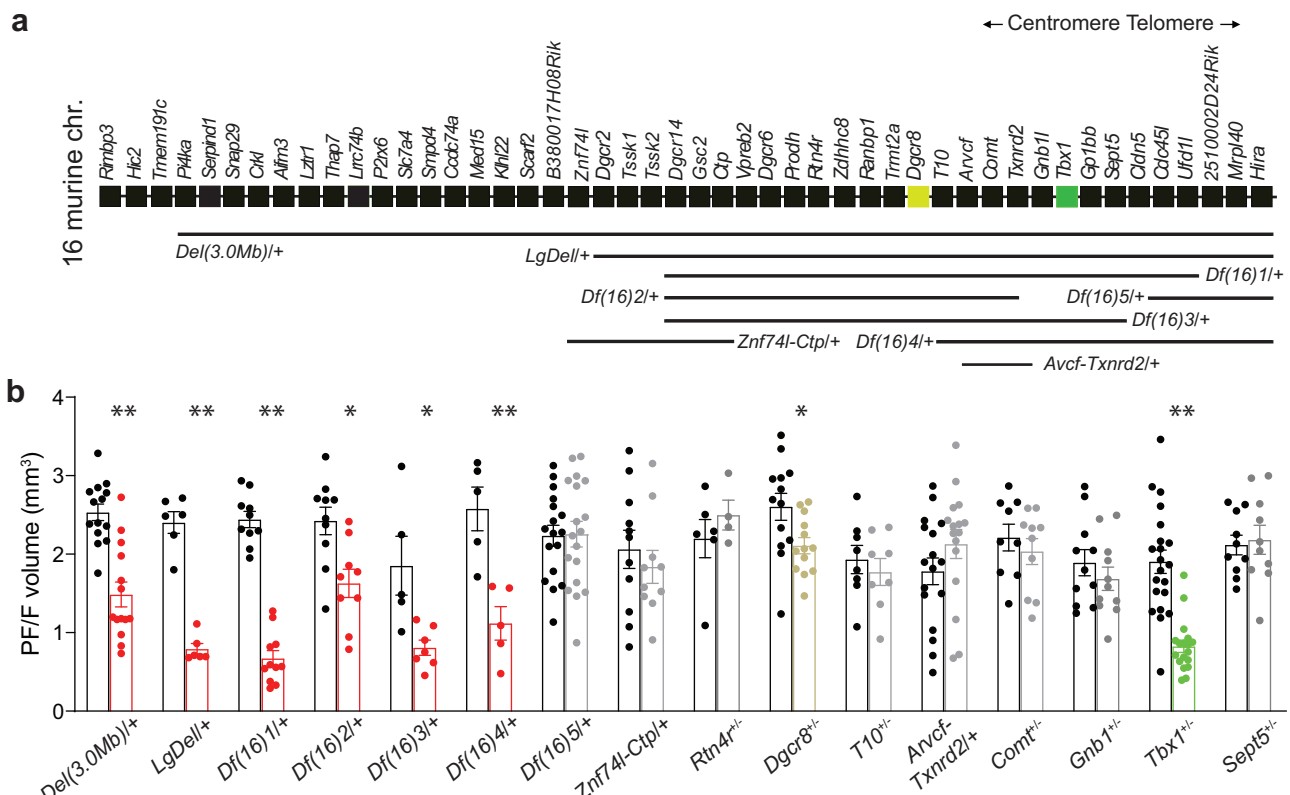

**Fig. 2 | Unbiased screen for the gene(s) responsible for paraflocculus/flocculus dysplasia in 22q11DS mice. a** Diagram depicting genes in the 22q11.2 syntenic region of mouse chromosome 16. Rectangles represent individual genes. Horizontal bars represent genomic regions hemizygously deleted in various mouse models of 22q11DS. **b** Mean PF/F volumes manually measured from MRI data of mice with hemizygous microdeletions of various sizes or hemizygous deletions of individual genes within the 22q11.2 syntenic region. Two-tailed Student's $t$-test (Holm–Sidak post hoc): $Del(3.0\,Mb)/+$ (red, 14 mice; 14 WT mice): $t_{26} = 5.47$, $^{**}p < 0.0001$; $LgDel/+$ (red, 6 mice; 6 WT mice): $t_{14} = 11.47$, $^{**}p < 0.0001$; $Df(16)1/+$ (red, 11 mice; 10 WT mice): $t_{19} = 12.39$, $^{**}p < 0.0001$; $Df(16)2/+$ (red, 9 mice; 10 WT mice): $t_{17} = 3.15$, $^{*}p = 0.006$; $Df(16)3/+$ (red, 7 mice; 5 WT mice): $t_{10} = 3.15$, $^{*}p = 0.01$; $Df(16)4/+$ (red, 5 mice; 5 WT mice): $t_8 = 4.17$, $^{**}p = 0.003$; $Df(16)5/+$ (gray, 18 mice; 17 WT mice): $t_{33} = 0.09$, $p = 0.93$; $Znf74l\text{-}Ctp/+$ (gray, 10 mice; 11 WT mice): $t_{19} = 0.69$, $p = 0.49$; $Rtn4r^{+/-}$ (gray, 4 mice; 6 WT mice): $t_8 = 0.88$, $p = 0.4$; $Dgcr8^{+/-}$ (gold, 13 mice; 13 WT mice): $t_{24} = 2.49$, $^{*}p = 0.02$; for $T10^{+/-}$ (gray, 8 mice; 8 WT mice): $t_{14} = 0.64$, $p = 0.53$; $Arvcf\text{-}Txnrd2/+$ (gray, 16 mice; 17 WT mice): $t_{31} = 1.38$, $p = 0.18$; $Comt^{+/-}$ (gray, 10 mice; 9 WT mice): $t_{17} = 0.74$, $p = 0.47$; $Gnb1l^{+/-}$ (gray, 11 mice; 11 WT mice): $t_{20} = 0.92$, $p = 0.37$; $Tbx1^{+/-}$ (green, 18 mice; 21 WT mice): $t_{37} = 6.15$, $^{**}p < 0.0001$; $Sept5^{+/-}$ (gray, 10 mice; 10 WT mice): $t_{18} = 0.30$, $p = 0.77$. Data are presented as the mean ± SEM. Source data are provided as a Source Data file.

$Gnb1l^{+/-}$ mice, $Comt^{+/-}$ mice, $Tbx1^{+/-}$ mice, and $Sept5^{+/-}$ mice; and their respective WT littermates. We did not test the $Gp1bb$ gene, as its primary function affects platelets[40,41]. Among those tested, only $Tbx1^{+/-}$ mice showed PF/F reduction compared to WT mice (Fig. 2b). This PF/F dysplasia was robust (56.6% ± 17.4% reduction) and comparable to that observed in $Df(16)3/+$ mice (56.4% ± 13.9% reduction) and $Df(16)1/+$ mice (72.4% ± 13.5% reduction). To further confirm that $Tbx1$ haploinsufficiency is a major contributor to PF/F dysplasia in 22q11DS mice, we tested if the PF/F reduction in $Tbx1^{+/-}$ mice differs from that in mutants with multigene deletions containing the $Tbx1$ gene by using a two-sample $t$-test. The reduction of the PF/F volume in $Tbx1^{+/-}$ mice was in line with differences in $Del(3.0\,Mb)/+$ ($t_{48.59} = -0.13$, $p = 0.9$), $Df(16)3/+$ ($t_{6.35} = -0.08$, $p = 0.94$), and $Df(16)4/+$ ($t_{11.12} = 0.99$, $p = 0.34$) mice, though we detected a significant difference from that in $LgDel/+$ ($t_{25.98} = 2.34$, $p = 0.03$) and $Df(16)1/+$ mice ($t_{48.62} = 3.15$, $p = 0.003$). The differences observed in $LgDel/+$ mice and $Df(16)1/+$ mice was due to variability in WT littermate controls, as the mean PF/F volumes in these mutant samples were not significantly smaller than that in $Tbx1^{+/-}$ mice [two-sample $t$-test; $LgDel/+$ ($t_{17.43} = 0.31$, $p = 0.76$) and $Df(16)1/+$ ($t_{21.37} = 1.21$, $p = 0.24$)]. The PF/F dysplasia in $Tbx1^{+/-}$ mice was more severe than in mice with the $Df(16)2/+$ deletion, which does not contain the $Tbx1$ gene ($t_{12.99} = 2.79$, $p = 0.02$). The mean PF/F volumes in $Tbx1^{+/-}$ mice were significantly smaller than that in $Df(16)2/+$ mice ($t_{11.07} = 4.07$, $p = 0.002$).

To search for the smaller-effect gene within the $Df(16)2$ region, we used a candidate gene approach and compared the PF/F volumes in $Rtn4^{+/-}$ mice, $Dgcr8^{+/-}$ mice, and their respective WT littermates. $Dgcr8^{+/-}$ mice but not $Rtn4^{+/-}$ mice showed mild PF/F dysplasia (19.1 ± 14.1% reduction), suggesting that $Dgcr8$ haploinsufficiency contributes to the small PF/F reduction in $Df(16)2/+$ mice. The severity of the PF/F reduction was not exacerbated in $Tbx1^{+/-};Dgcr8^{+/-}$ mice compared to that in $Tbx1^{+/-}$ mice, indicating that there is no additive effect of haploinsufficiency of these two genes (Supplementary Fig. 3c). Similar to that in $Df(16)1/+$ mice, the PF/F dysplasia in $Tbx1^{+/-}$ mice did not depend on sex or laterality and occurred in the absence of changes in the total-brain and cerebellar volumes (Supplementary Fig. 3d–g). Furthermore, at the regional level in the cerebellum, the phenotype of $Tbx1^{+/-}$ mice was similar to that of $Df(16)1/+$ mice, as no differences were detected in the Crus I, Crus II, or Vermis IV/V volumes in the same mice that exhibited a robust reduction in the PF/F volume compared to WT littermates (Supplementary Fig. 3h). Together, these data suggest that $Tbx1$ haploinsufficiency is a major contributor to the PF/F dysplasia in 22q11DS mouse models.

## Cerebellar-dependent motor learning is impaired in 22q11DS mice

To test for functional consequences of the hampered PF/F structural development in 22q11DS, we examined the adaptation of the

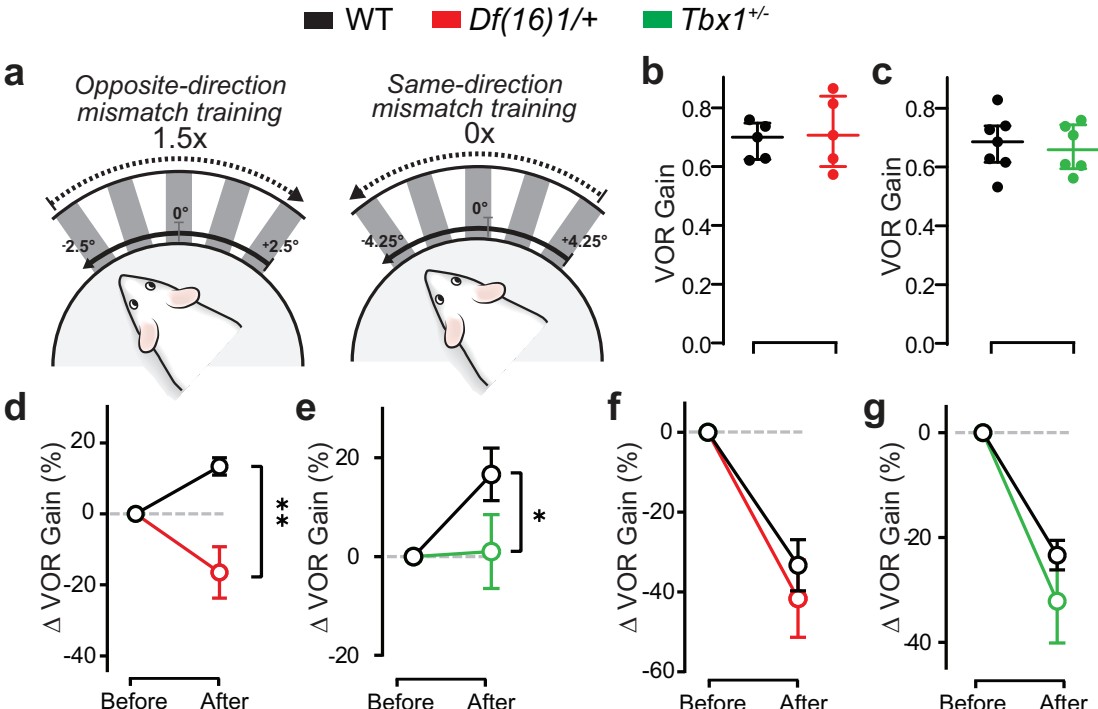

**Fig. 3 | Compromised vestibulo-ocular reflex–based motor learning and cerebellar-dependent plasticity in 22q11DS mice. a** Diagram of the experimental setup for measuring the vestibulo-ocular reflex (VOR) in head-fixed mice on a rotating platform with a visual grating moving in the opposite or same direction, relative to the head[169]. **b, c** The basal level of VOR gain in *Df(16)1/+* mice (red, *n* = 5) and *Tbx1*[+/−] mice (green, *n* = 6) and their respective WT controls (black, **b** 5 mice, **c** 7 mice) measured immediately before training. Two-tailed unpaired Student's *t*-test: **b** $t_8$ = 0.46, *p* = 0.66; **c** $t_{11}$ = 0.32, *p* = 0.76. **d, e** Average VOR gain after opposite-

direction mismatch training (1.5×, 60 min) in WT (**d** 5 mice, **e** 7 mice), *Df(16)1/+* (*n* = 5), and *Tbx1*[+/−] (*n* = 6) mice. Two-way ANOVA (Holm–Sidak's post hoc) after training: **d** $F_{1,8}$ = 15.3, \*\**p* < 0.0001; **e** $F_{1,11}$ = 3.03, \**p* = 0.04. **f, g** Average VOR gain after same-direction visual-vestibular mismatches (0×, 60 min) in WT (**f** 5 mice, **g** 6 mice), *Df(16)1/+* (*n* = 4), and *Tbx1*[+/−] (*n* = 5) mice. Two-way ANOVA (Holm–Sidak's post hoc) after training: **f** $F_{1,7}$ = 0.56, *p* = 0.52; **g** $F_{1,9}$ = 1.26, *p* = 0.24. Data are presented as the mean ± SEM. Source data are provided as a Source Data file.

vestibulo-ocular reflex (VOR), a type of motor learning that involves the PF/F[42]. To assess VOR performance, we head-fixed mice on a rotating platform and used video-oculography to measure eye movements evoked by sinusoidal vestibular stimuli. Compared to WT littermates, *Df(16)1/+* and *Tbx1*[+/−] mice (older than P160) showed a similar baseline VOR performance (1 Hz; 0.69 ± 0.06 and 0.72 ± 0.12 for WT and *Df(16)1/+* mice, respectively, and 0.68 ± 0.1 and 0.66 ± 0.08 for WT and *Tbx1*[+/−] mice, respectively) (Fig. 3b, c). These results suggested that vestibular-induced oculomotor control is not impaired in these animals. Interestingly, *Df(16)1/+* mice exhibited clear cerebellar-dependent motor-learning abnormalities. After opposite-direction visual-vestibular mismatch training (1.5×; 60 min, Fig. 3a), *Df(16)1/+* mice adapted their VOR aberrantly: the gain (or amplitude) of their response decreased instead of increasing (Fig. 3d), a response that would exacerbate the apparent motor error instead of correcting it. Yet, in response to same-direction mismatch training (0×; 60 min, Fig. 3a) in a separate session, these same mice adapted their VOR in the correct direction. The amplitude of gain-decrease VOR adaptation was the same as that induced in WT control mice (Fig. 3f). This result indicates a context-dependent effect of the 22q11.2 microdeletion on cerebellum-dependent motor learning. We also tested VOR after opposite- and same-direction mismatch training in *Tbx1*[+/−] mice and found a deficit of VOR gain for the opposite-direction training context, though to a lesser extent than that in *Df(16)1/+* mice (Fig. 3e), but normal VOR adaptation in the same-direction training context (Fig. 3g). These results suggest that VOR adaptation is disrupted in a context-dependent manner in the 22q11DS mouse model, and *Tbx1* haploinsufficiency contributes to this deficit.

## Cerebellar-dependent plasticity is compromised in the PF/F of 22q11DS mice

VOR adaptation depends on different cerebellar-dependent plasticity mechanisms to increase or decrease the response gain[43,44]. Gain-increase learning (induced by opposite-direction visual-vestibular motion mismatches) depends on climbing fiber (Cf)-induced long-term depression (LTD) at parallel fiber (Pf)–Purkinje cell (PC) synapses[45,46]. Gain-decrease learning (same-direction visual-vestibular motion mismatches) relies on long-term potentiation (LTP) at Pf–PC synapses and is induced independently of Cf activity[47]. Thus, we next tested if disruption of the VOR gain-increase shown in *Df(16)1/+* and *Tbx1*[+/−] mice is correlated with abnormal Pf–PC LTD/LTP in the PF/F (Fig. 4a–j).

We measured LTD and LTP in acute brain slices containing the PF/F. We induced LTD at PF/F Pf–PC synapses by conjunctive synaptic stimulation of Pf and Cf inputs (Fig. 4a). Compared to baseline, conjunctive stimulation reduced the amplitude of excitatory postsynaptic potentials (EPSPs) by about 50% in WT mice. In contrast, the same induction protocol in *Df(16)1/+* mice potentiated the amplitude of EPSPs at Pf–PC synapses instead of inducing LTD (Fig. 4b, c). Similar to *Df(16)1/+* mice, *Tbx1*[+/−] mice also showed defective LTD compared to their WT littermates (Fig. 4d, e). The Cf inputs to PCs change their strength and numbers during development[48], and these processes could be affected in 22q11DS. However, we observed no differences in the baseline Cf or Pf synaptic responses in *Df(16)1/+* mice or *Tbx1*[+/−] mice (Supplementary Fig. 4), suggesting that these synaptic inputs to PCs are grossly normal. We then compared Pf–PC LTP induced by Pf stimulation without Cf co-stimulation (Fig. 4f). In contrast to LTD, the LTP in *Df(16)1/+* mice was similar to that in WT mice (Fig. 4g, h). *Tbx1*[+/−]

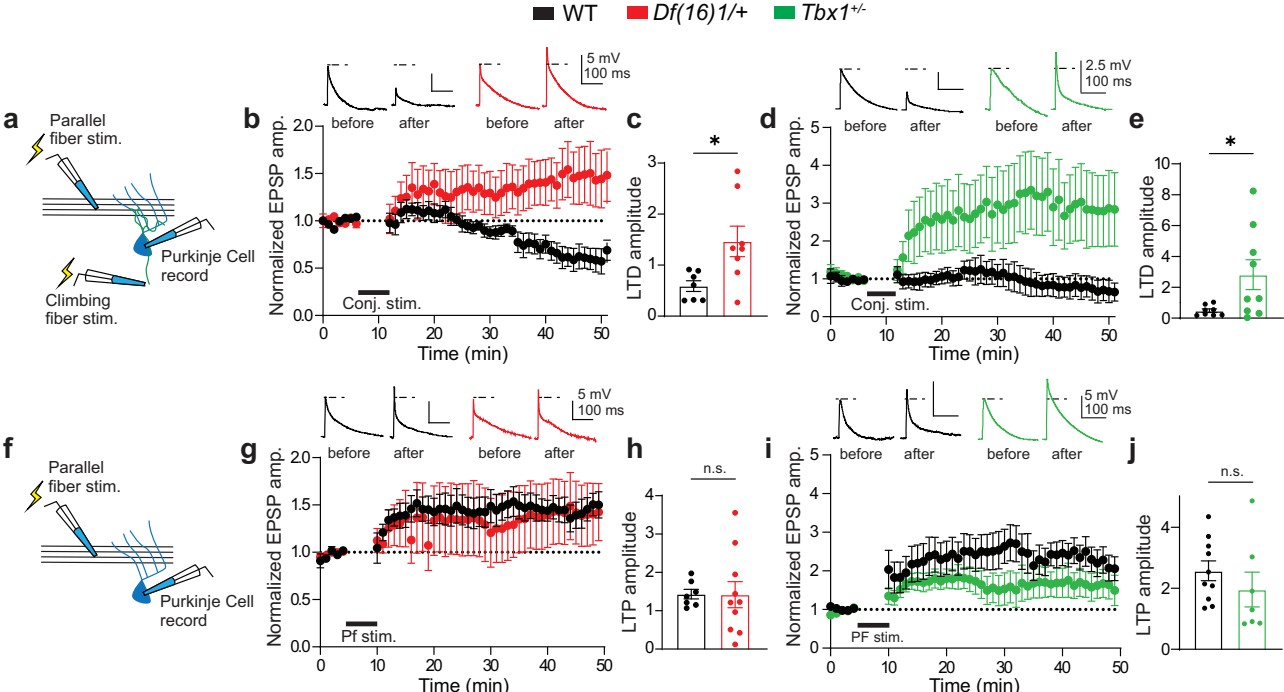

**Fig. 4 | Compromised cerebellar-dependent plasticity in the paraflocculus/ flocculus in 22q11DS mice. a** Schematic of the recording and stimulation configuration for long-term depression (LTD) induction. Purkinje cells (PCs) were targeted for whole-cell recording while parallel fiber (Pf) and climbing fiber (Cf) inputs were stimulated. **b** Normalized excitatory postsynaptic potential amplitude (EPSP amp.) shows the LTD time course before and after a 10-min baseline in WT (black) and *Df(16)1/+* (red) PCs; LTD was induced by conjunctive Pf and Cf stimulation. Example voltage traces from individual cells (upper) show EPSPs induced by Pf stimulation before (left) and after (right) LTD induction in WT and *Df(16)1/+* PCs. The dotted lines show the amplitude of the baseline response. **c** Average LTD amplitude in WT (n = 7 cells, 4 mice) and *Df(16)1/+* (n = 8 cells, 5 mice) PCs. Two-tailed Student's *t*-test (Welch's correction): $t_{8.66} = 2.77$, *$p = 0.023$. **d** Normalized EPSP amplitude shows the LTD time course, before and after the 10-min baseline in WT (black) and *Tbx1+/-* (green) PCs. Example voltage traces from individual cells (upper) show EPSPs induced by Pf stimulation before (left) and after (right) LTD

induction in WT and *Tbx1+/-* PCs. The dotted lines show the amplitude of the baseline response. **e** Average LTD amplitude in WT (n = 8 cells, 5 mice) and *Tbx1+/-* (n = 9 cells, 6 mice) PCs. Two-tailed Student's *t*-test (Welch's correction): $t_{8.24} = 2.41$, *$p = 0.042$. **f** Schematic of the recording and stimulation configuration for long-term potentiation (LTP) induction. PCs were targeted for whole-cell recording while Pf inputs were stimulated. **g** Normalized LTP time course before and after 5-min baseline in WT and *Df(16)1/+* PCs, showing LTP induction by tetanic PF stimulation. **h** Average LTP amplitude in WT (n = 7 cells, 5 mice) and *Df1+/-* (n = 10 cells, 5 mice) PCs. Two-tailed Student's *t*-test (Welch's correction): $t_{11.33} = 0.04$, $p = 0.968$. **i** Normalized the LTP time course before and after 5-min baseline in WT and *Tbx1+/-* PCs, showing LTP induction by tetanic PF stimulation. **j** Average LTP amplitude in WT (n = 10 cells, 8 mice) and *Tbx1+/-* (n = 7 cells, 5 mice) PCs. Two-tailed Student's *t*-test (Welch's correction): $t_{9.86} = 0.94$, $p = 0.37$. Data are presented as the mean ± SEM. n.s. not significant. Source data are provided as a Source Data file.

mice also showed no difference in LTP compared to their matched littermates (Fig. 4i, j). These data suggest that *Tbx1* haploinsufficiency compromises Pf–PC LTD but not LTP in the PF/F of 22q11DS mice, aligning with the context dependence of VOR learning deficits in these mouse models.

### Cerebellar composition and neurogenesis are normal in the PF/F of 22q11DS mice

We next sought to identify the cellular mechanisms of the PF/F dysplasia in 22q11DS. We examined cell proliferation, apoptosis, and migration of the cerebellar granule neuron precursors (CGNPs), which expand in the external granular layer (EGL) and exit the cell cycle to become differentiated cerebellar granule neurons (CGNs), which constitute the vast majority of cells in the cerebellum[49]. We investigated whether PF/F dysplasia in 22q11DS mice was associated with a reduced rate of proliferation or an increased rate of apoptosis. Cerebellar tissues from mice (aged P3–P14) were probed with Ki67 and phosphohistone H3 (PH3), markers of proliferation, and cleaved caspase-3, a marker of apoptosis (Fig. 5a–i). Interestingly, 22q11DS did not affect the proliferation or apoptosis of CGNPs in the PF/F. The density of Ki67+ cells and PH3+ cells in the EGL of the PF/F of *Df(16)1/+* mice did not differ from that in WT littermates during P3 to P7 (Fig. 5a–f). Also, we detected no change in apoptosis (cleaved caspase-3+ cells) during P3 to P14 between *Df(16)1/+* mice and WT littermates (Fig. 5g–i).

We next examined if the exit of postmitotic CGNs from the germinal zone and migration to the inner cerebellar layer is affected in 22q11DS mice. We used ex vivo organotypic cerebellar slices that were electroporated with an H2B-mCherry–labeled expression construct and maintained in culture for 48 h to analyze the distribution of CGN migration in the PF of *Df(16)1/+* mice and WT mice. CGN migration did not differ between *Df(16)1/+* mice and WT littermates (Fig. 5j, k). Furthermore, we detected no difference in the cellular composition of the PF/F between 22q11DS and WT mice. The density of PCs (calbindin+ cells) in the PF/F of P5 to P14 *Df(16)1/+* mice did not differ from that of WT littermates (Supplementary Fig. 5a–g). Also, there was no difference in the density of CGNPs (Pax6+), astrocytes (GFAP+) (Supplementary Fig. 5h, i), or unipolar brush cells (Tbr2+), relative to that of neurons (beta-III tubulin+) in the PF/F, between *Df(16)1/+* and WT littermates (Supplementary Fig. 5j–l). These results suggest that PF/F dysplasia is not caused by intrinsic changes in cerebellar composition or neurogenesis within the PF/F in 22q11DS mice.

### *Tbx1* haploinsufficiency is associated with malformations of the subarcuate fossa and semicircular canals

The PF differs from other cerebellar lobules in that it is encapsulated by part of the petrous temporal bone (PTB) called the subarcuate fossa (SF)[34,50]. Therefore, we next tested whether 22q11DS affects the neurocranium surrounding the PF. Computed tomography (CT) analysis

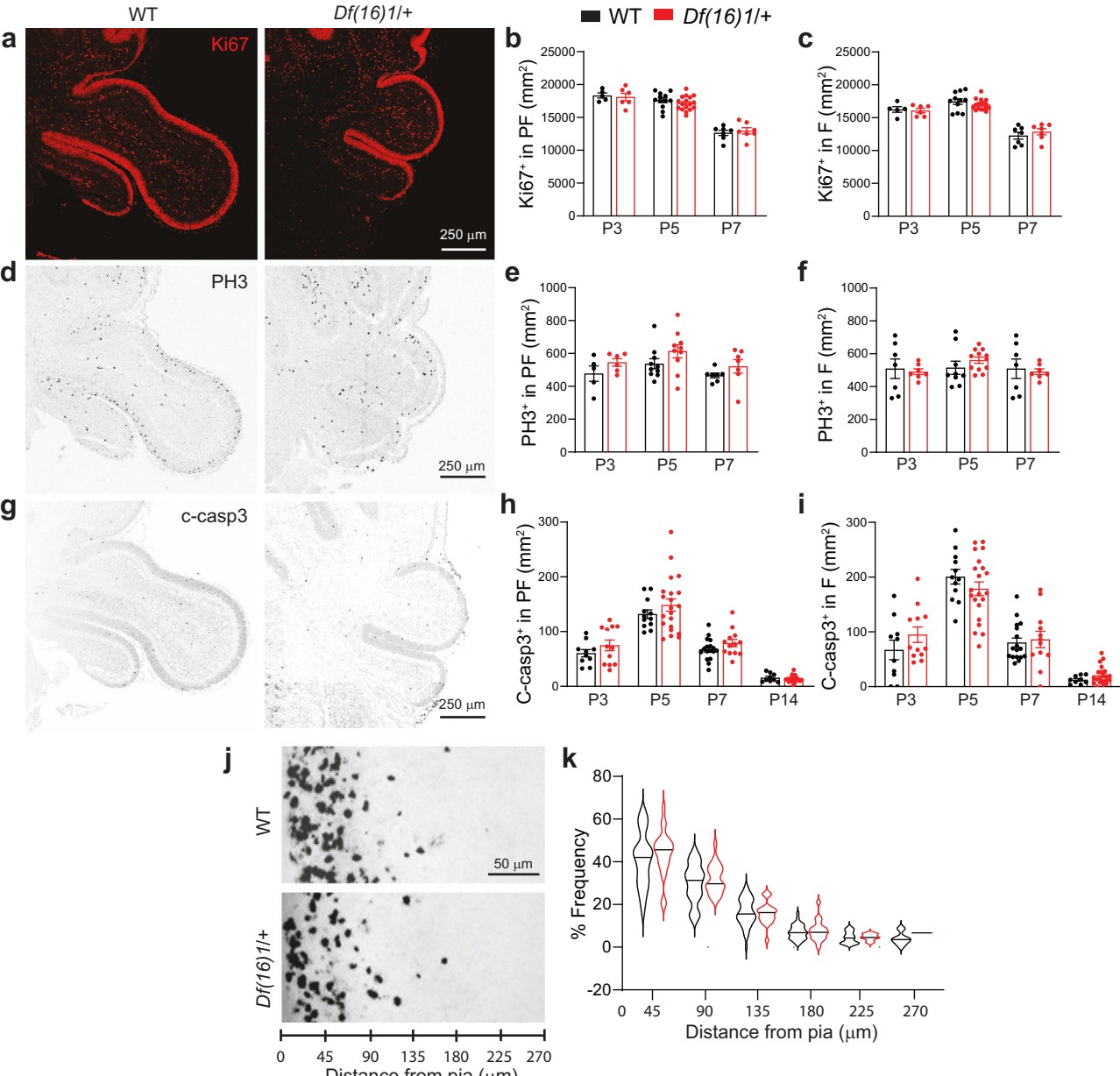

**Fig. 5 | Cerebellar progenitor neurogenesis and migration are normal in the paraflocculus/flocculus of 22q11DS mice. a**, **d** Representative confocal images of Ki67 and PH3 (proliferation markers, red and black, respectively) staining in the external germinal layer (EGL) of P7 WT (black) and *Df(16)1/+* (red) mice.
**b, c, e, f** Average normalized numbers of Ki67⁺ cells (**b** WT: 24 sections, 9 mice; *Df(16)1/+*: 30 sections, 12 mice; **c** WT: 23 sections, 9 mice; *Df(16)1/+*: 30 sections, 12 mice) and PH3⁺ cells (**e** WT: 22 sections, 9 mice; *Df(16)1/+* mice: 23 sections, 12 mice; **f** WT: 23 sections, 9 mice; *Df(16)1/+*: 26 sections, 12 mice) in the PF or F of P3–P7 WT and *Df(16)1/+* animals. Two-way ANOVA (Holm–Sidak's post hoc); **b** $F_{1,50} = 0.42$, $p = 0.52$ (P3: $p = 0.89$, P5: $p = 0.89$, P7: $p = 0.89$); **c** $F_{1,47} = 0.005$, $p = 0.95$ (P3: $p = 0.79$; P5: $p = 0.6$; P7: $p = 0.6$); **e** $F_{1,39} = 5.0$, $p = 0.03$ (P3: $p = 0.44$; P5: $p = 0.24$; P7: $p = 0.44$); **f** $F_{1,43} = 0.009$, $p = 0.93$ (P3: $p = 0.93$; P5: $p = 0.7$; P7: $p = 0.93$). **g** Representative confocal images of cleaved caspase-3 (c-casp3, apoptosis marker, black) in the PF/F of P7 WT and *Df(16)1/+* mice. **h, i** Average normalized numbers of c-casp3⁺ cells in

the PF or F of P3–P14 WT (**h, i** 48 sections, 16 mice) and *Df(16)1/+* (**h** 68 sections, 20 mice; **i** 66 sections, 20 mice) animals. Two-way ANOVA (Holm–Sidak's post hoc); **h** $F_{1,108} = 3.17$, $p = 0.08$ (P3: $p = 0.58$; P5: $p = 0.39$; P7: $p = 0.58$; P14: $p = 0.86$); **i** $F_{1,106} = 0.34$, $p = 0.56$ (P3: $p = 0.44$; P5: $p = 0.44$; P7: $p = 0.87$; P14: $p = 0.87$). **j, k** CGN migration in the PF is normal in *Df(16)1/+* animals. **j** Representative confocal images of cerebellar slices showing H2B-mCherry–labeled CGN germinal zone exit and migration. **k** Cellular distribution analyzed in each binned distance in P7 WT (27 brain sections, 7 mice) and *Df(16)1/+* (8 brain sections, 5 mice) animals. Two-way ANOVA (Holm–Sidak's post hoc); $F_{51,288} = 0.71$, $p = 0.4$ (0–45 μm: $p = 0.6$; 45–90 μm: $p = 0.85$; 90–135 μm: $p = 0.99$; 135–180 μm: $p = 0.99$; 180–225 μm: $p = 0.99$; 225–270 μm: $p = 0.99$). Data are presented as the mean ± SEM, except in (**k**), where data are presented as a violin plot with median values. Source data are provided as a Source Data file.

showed a significant reduction in the SF depth and volumes at different ages in *Df(16)1/+* and *Tbx1*⁺/⁻ mice compared to WT littermates (Fig. 6a–f). The three-dimensional (3D) reconstruction showed that the SF space housing the PF was substantially smaller in *Df(16)1/+* mice (Fig. 6a, b, e) and *Tbx1*⁺/⁻ mice (Fig. 6c, d, f) compared to that in WT littermates. At different angles, 2D imaging revealed a substantial

reduction in the SF space in *Df(16)1/+* mice compared to that in WT littermates, and this effect was detected from P7 to adulthood (2–3 months) (Fig. 6a'–a''', b'–b''', e). Similar results were obtained in *Tbx1*⁺/⁻ mice; the SF volume was substantially smaller in *Tbx1*⁺/⁻ mice than in WT littermates from P7 to adulthood (2–3 months) (Fig. 6c'–c''', d'–d''', f).

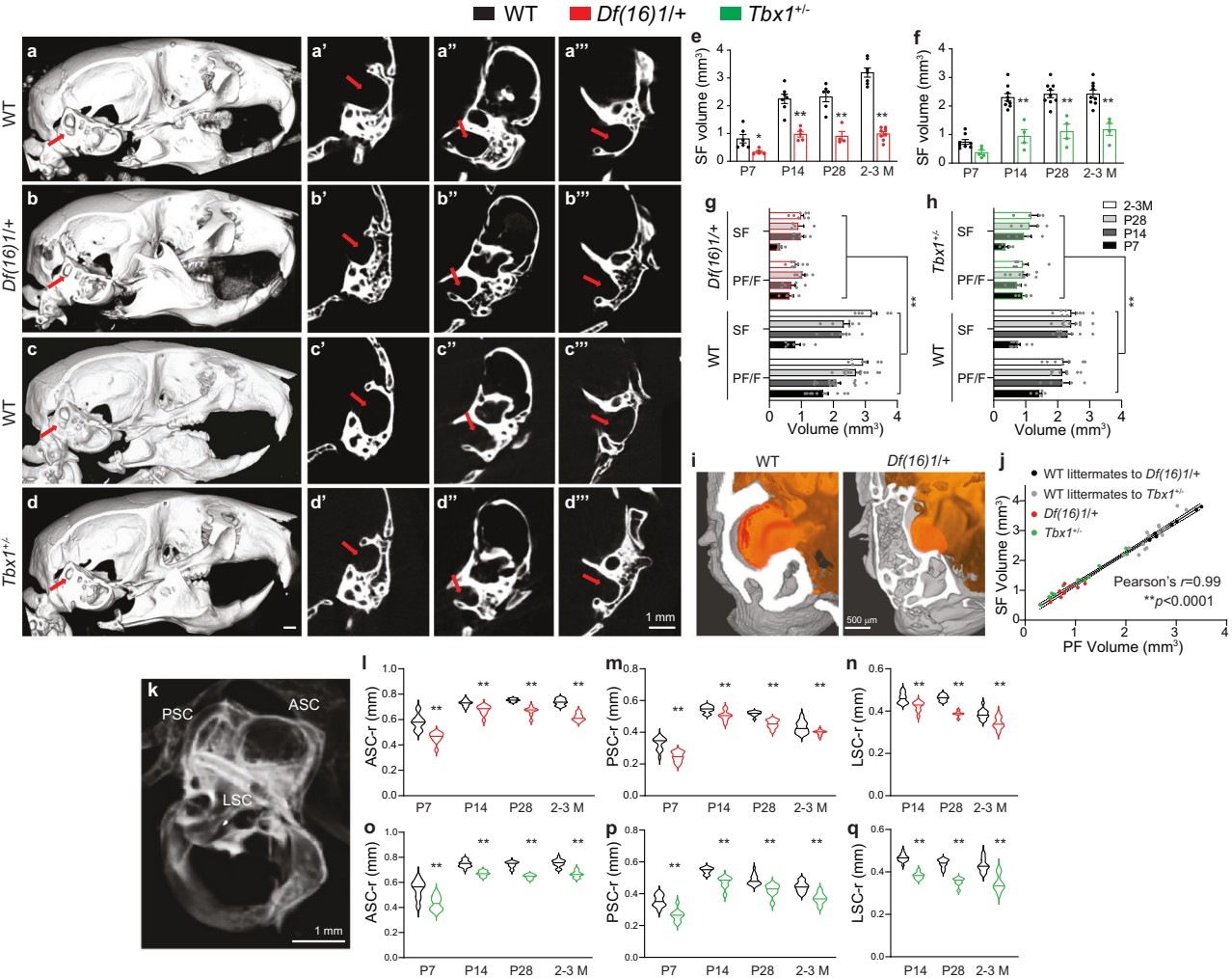

**Fig. 6 | Skeletal defects in the subarcuate fossa and semicircular canals in 22q11DS mice and *Tbx1*⁺ᐟ⁻ mice.** **a–h** Representative CT scans of the skull in 3D lateral (**a–d**), 2D coronal (**a′–d′**), axial (**a″–d″**), and sagittal (**a‴–d‴**) orientations, with the subarcuate fossa (SF, arrows) shown in 2-month-old *Df(16)1/+* (red) and *Tbx1*⁺ᐟ⁻ (green) mice with their respective WT controls. **e, f** Average volumes of the SF in the *Df(16)1/+* (**e** 22 mice), *Tbx1*⁺ᐟ⁻ mice (**f** 17 mice), and their respective WT controls (**e** 26 mice, **f** 35 mice) at different ages. Two-way ANOVA (Holm–Sidak's post hoc); **e** $F_{1,40}$ = 166.5; **p* < 0.0001 (P7: **p* = 0.03; P14 to 2–3 M: **p* < 0.0001); **f** $F_{1,44}$ = 93.53; **p* < 0.0001 (P7: *p* = 0.09; P14 to 2–3 M: **p* < 0.0001). **g, h** Similar volumetric decrease of the SF and PF/F during development (aged P7 to 2–3 M) in *Df(16)1/+* (**g** 46 mice) and *Tbx1*⁺ᐟ⁻ mice (**h** 39 mice) compared to WT controls (**g** 71 mice; **h** 62 mice). Three-way ANOVA; **g** WT vs *Df(16)1/+* *F* = 412.74, **p* < 0.0001; **h** WT vs T*bx1*⁺ᐟ⁻ *F* = 198.04, **p* < 0.0001. **i** 3D visualization of CT and ex vivo MRIs of

the SF and PF/F (related to Supplementary Movies 3, 4) in WT and *Df(16)1/+* mice. **j** Linear regression in 2-month-old *Df(16)1/+* (*n* = 8), *Tbx1*⁺ᐟ⁻ (*n* = 13), and age-matched WT (*n* = 7 and 15, respectively) mice. Pearson's *r* = 0.99, **p* < 0.0001. **k** CT image of the bony labyrinth, including the anterior, posterior, and lateral SCs (ASC, PSC, LSC, respectively). **l–q** Radius arc (-r) of the ASCs (**l, o**), PSCs (**m, p**), and LSCs (**n, q**) in *Df(16)1/+* mice (*n* = **l** 22, **m** 22, **n** 16), *Tbx1*⁺ᐟ⁻ mice (*n* = **o** 17, **p** 17, **q** 12) and their WT controls (*n* = **l** 27, **m** 27, **n** 20, **o** 35, **p** 35, **q** 27) at different ages. Two-way ANOVA (Holm–Sidak's post hoc); **l** $F_{1,90}$ = 170.6; **p* < 0.0001; **m** $F_{1,89}$ = 69.68; **p* < 0.0001; **n** $F_{1,66}$ = 87.32; **p* < 0.0001; **o** $F_{1,96}$ = 121.1; **p* < 0.0001; **p** $F_{1,95}$ = 118; **p* < 0.0001; **q** $F_{1,71}$ = 194.7; **p* < 0.0001. Data are presented as the mean ± SEM in (**e–h**) and as violin plots with median values in (**l–q**). Source data are provided as a Source Data file.

The SF abnormality in *Df(16)1/+* and *Tbx1*⁺ᐟ⁻ mice was not a result of global skeletal deformation, as the total-skull volume was not affected in either strain, compared to that in their respective WT controls (Supplementary Fig. 6a, b). However, the volume of the PTB encompassing the SF was significantly smaller in *Df(16)1/+* and *Tbx1*⁺ᐟ⁻ mice than in WT mice (Supplementary Fig. 6c–f). We next examined the volumetric changes of SF and PF during development in *Df(16)1/+* mice (Fig. 6f) and *Tbx1*⁺ᐟ⁻ mice (Fig. 6h), compared to matched littermates. The volumes of SF and PF in control mice showed similar growth extensions as they developed, which was obstructed in both mutant mice. A combined CT and MRI 3D visualization showed obstructed occupancy of the PF within the SF in *Df(16)1/+* mice compared to WT mice (Fig. 6i and Supplementary Movies 3, 4). Consistent with the notion that the SF houses the PF and their development is interlinked,

a regression analysis showed that the SF volume was strongly correlated with the PF volume (Fig. 6j; Pearson's *r* = 0.99, *p* < 0.0001).

The SF is spatially interlinked with the semicircular canals (SCs) of the inner ear, comprising the anterior, posterior, and lateral semicircular canals (ASC, PSC, and LSC, respectively) (Fig. 6k). Of these three canals, the ASC is the most proximal to the opening of the SF into the endocranial cavity, and the sizes of these two structures are correlated during development[51]. The ASC, PSC, and LSC were diminished in *Df(16)1/+* mice and *Tbx1*⁺ᐟ⁻ mice, relative to their WT counterparts at all tested ages (P7 to 2–3 months; Fig. 6l–q). There was also a significant decrease in the heights and widths of all three canals in *Df(16)1/+* mice and *Tbx1*⁺ᐟ⁻ mice at all tested ages (Supplementary Fig. 6j–u). We detected a malformation of the LSC in 9.1% of the *Df(16)1/+* mice and 3.6% of the *Tbx1*⁺ᐟ⁻ mice; thus, measurements in

those mice were omitted. We calculated the radius of each canal based on the height and width data, and all three SCs were significantly smaller at all ages in *Df(16)1/+* mice (Fig. 6l–n) and *Tbx1⁺ᐟ⁻* mice (Supplementary Fig. 6o–q), compared to their respective WT counterparts. Moreover, the degree of SC decrease was similar between the two mutant strains. These results suggest that *Tbx1* haploinsufficiency in *22q11DS* mice specifically disrupts the normal development of the skeletal structures (SF and SCs) that encapsulate or are closely connected to the PF/F.

### *Tbx1* haploinsufficiency leads to an abnormal chondrocyte-to-osteoblast transition in the PTB

To further elucidate the effects of *Tbx1* haploinsufficiency on cell-type composition and shifts in cell type–specific transcriptomes in the PTB and PF/F, we performed single-nuclei RNA sequencing (snRNA-seq) analysis of the PF/F and surrounding bone tissues in P5.5 WT and *Tbx1⁺ᐟ⁻* mice. We performed this analysis in four groups of mice, consisting of two separate batches of animals of each genotype. After removing low-quality nuclei, we used 45,914 cells for in-depth analysis of the bone and 48,237 cells for in-depth analysis of the PF/F tissues.

Louvain clustering integration of the WT and *Tbx1⁺ᐟ⁻* sample datasets indicated that all defined cell populations were maintained across both genotypes (Supplementary Fig. 7a, b). Subsequent cell-type annotation revealed a complex landscape in the bone atlas, encompassing a range of subtypes of progenitors, connective tissue (CT; also called cartilage), multiple chondrocyte progenitors (CPs), osteoblasts/osteocytes, immune cells, and neurons, suggesting cellular heterogeneity in the bone cell population (Supplementary Fig. 7c). Among the sequenced cells, most (68%) comprised CT, chondrocytes, and osteoblasts (Supplementary Fig. 7c). *Tbx1* expression was pronounced in these major cell populations in WT mice, especially in CP subtypes (Supplementary Fig. 7d), and it was significantly reduced in *Tbx1⁺ᐟ⁻* mice (Supplementary Fig. 7e). Thus, we focused our downstream analysis on these cell populations.

Uniform manifold approximation and projection (UMAP) analysis based on the expression of marker genes showed four subtypes of CT (CT1-4), six subtypes of CPs (CP1-6), and differential states of chondrocyte and osteoblast populations (hypertrophic chondrocytes, osteoblasts, matured osteoblasts, and osteoclasts) (Fig. 7a). Marker gene expression analysis within the chondrocyte–osteoblast populations highlighted distinct, overlapping patterns and provided insights into their developmental connectivity (Fig. 7b). Marker genes for CT, such as *Col3a1*, which encodes a component of type III collagen, and *Vcan*, which encodes the large extracellular matrix proteoglycan versican commonly present in CT and CP subtypes, suggested a lineage relationship between these cell types based on a shared developmental pathway or a transitional differentiation state from CT to CPs. The chondrocyte marker gene *Col2a1*, which encodes the alpha-1 chain of type II collagen, a major component of cartilage extracellular matrix, was expressed across all chondrocytes and most of the osteoblast subtypes. This expression pattern indicated a commonality in the extracellular matrix composition between these cell types, despite their specialized functions, and supports the notion that chondrocytes differentiate into osteoblasts via endochondral ossification processes[52,53]. Furthermore, the expression of osteoblast markers, including *Alpl* and *Bglap*, in hypertrophic chondrocytes and osteoblasts confirmed the progression of chondrocytes into an osteogenic phenotype, a critical step in bone development and homeostasis. Osteoclasts, which are distinct from the osteogenic lineage, showed a unique expression profile, with markers *Acp5* and *Ctsk* reflecting their specialized role in bone resorption[54].

UMAP analysis of the annotated cerebellar cell compositions in the PF/F adhered to the canonical GABAergic and glutamatergic lineages (Fig. 7c), which was consistent with the published cerebellar atlas[55]. Marker gene expression analysis further affirmed the classification of cells into their respective lineages (Fig. 7d). Marker genes indicative of CGNPs, such as *Sox6* and *Neurod1*, and *Pax2*, which is associated with the GABAergic lineage, were clearly delineated. *Tbx1* expression was not detected across cerebellar types in the postnatal PF/F. We detected *Tbx1* expression only in the developing WT mouse cerebellum, mainly in the rhombic lip at embryonic day (E) 10; *Tbx1* was expressed at negligible levels in other cerebellar cell types and postnatally (Supplementary Fig. 7f).

Based on these single-cell transcriptional landscapes in the PTB and PF/F, we next compared cell-type compositions in WT and *Tbx1⁺ᐟ⁻* mice (Fig. 7e–j and Supplementary Fig. 7g–j). *Tbx1⁺ᐟ⁻* mice showed a significant alteration in the cellular dynamics within the chondro-osteogenic lineage (Fig. 7e, f) and minor changes in other cell types (Supplementary Fig. 7g, h). The ratio of chondrocytes to osteoblasts consistently decreased, as supported by data from two separate batches (Fig. 7g). This included a significant reduction in CPs accompanied by a concurrent rise in the populations of hypertrophic chondrocytes, osteoblasts, and mature osteoblasts (Supplementary Fig. 7g, i). In contrast, cerebellar (PF/F) cell populations remained stable; no significant shifts (Fig. 7h–j and Supplementary Fig. 7h, j) and no change in cell proliferation, cell cycle, or apoptosis was observed between WT and *Tbx1⁺ᐟ⁻* mice (Supplementary Fig. 8). This disparity underscores the selective role of *Tbx1* in transitions of chondrocyte-to-osteoblast lineages during PTB development.

### Manipulating *Tbx1* levels fails to mimic or rescue PF/F dysplasia in 22q11DS mice

We next tested if *Tbx1* deletion in various lineages would affect the PF/F structure. Mice carrying the *Tbx1* floxed allele (*Tbx1ᶠˡ/⁺* mice[56]) were crossed with the following Cre-expressing mice: *Tie2ᶜʳᵉ* mice[57] (endothelial cells), *L7ᶜʳᵉ* mice[58] (PCs), *NeuroD1ᶜʳᵉ* mice (MGI5311739; CGNs), *Lyz2ᶜʳᵉ* mice[59] (osteoclasts), *Twist2ᶜʳᵉ* mice[60] (osteoprogenitor cells), *Mesp1ᶜʳᵉ* mice[61] (mesodermal cells), and *Foxg1ᶜʳᵉ* mice[62] (telencephalon and otic vesicle–expressed cells). Hemizygous or homozygous conditional *Tbx1* deletion in these Cre lines showed no effect on the PF/F volumes compared to those of WT littermates (Supplementary Fig. 9a–g). Notably, only hemizygous *Tbx1* deletion was assessed in *Mesp1ᶜʳᵉ*, *Twist2ᶜʳᵉ*, or *Foxg1ᶜʳᵉ* mice (Supplementary Fig. 9e–g), as homozygous mutants were not viable.

Our attempts to restore the PF/F volume in *Tbx1⁺ᐟ⁻* mice were also unsuccessful. Overexpression of *Tbx1* in the mesoderm, osteoprogenitor cells, or otic vesicle–expressing cells was embryonically lethal. To that end, we crossed the *Tbx1-GFPᶠˡ/⁺* line[63] with *Mesp1ᶜʳᵉ*, *Twist2ᶜʳᵉ*, or *Foxg1ᶜʳᵉ* mice, but *Mesp1ᶜʳᵉ;Tbx1-GFPᶠˡ/⁺* mice, *Twist2ᶜʳᵉ;Tbx1-GFPᶠˡ/⁺* mice, and *Foxg1ᶜʳᵉ;Tbx1-GFPᶠˡ/⁺* mice were not viable. Moreover, these mice crossed with *Tbx1⁺ᐟ⁻* mice (*Tbx1⁺ᐟ⁻;Mesp1ᶜʳᵉ;Tbx1-GFPᶠˡ/⁺* mice, *Tbx1⁺ᐟ⁻;Twist2ᶜʳᵉ;Tbx1-GFPᶠˡ/⁺* mice, or *Tbx1⁺ᐟ⁻;Foxg1ᶜʳᵉ;Tbx1-GFPᶠˡ/⁺* mice) also were not viable.

Previous works have inhibited chromatin modification or suppressed p53 expression to rescue deficiencies in the cardiac outflow tract in *Tbx1⁺ᐟ⁻* mice[64,65]. We intraperitoneally injected tranylcypromine, an inhibitor of histone demethylases, into pregnant mice at E7.5 to E15.5, the period including high expression of *Tbx1*[66], or at E11.5 to E18.5, the period when the SCs and primitive SF develop[50]. However, tranylcypromine did not rescue the PF/F volumes in P28 *Df(16)1/+* or *Tbx1⁺ᐟ⁻* mice (Supplementary Fig. 9h, i). Similarly, *p53⁺ᐟ⁻;Tbx1⁺ᐟ⁻* mice showed no rescue of the PF/F volumes (Supplementary Fig. 9j).

## Discussion

Mild-to-moderate structural changes in the cerebellum have been consistently observed in 22q11DS[28,67–71] and SCZ[72,73]. These alterations are thought to be connected to functional impairments, including cognitive deficits and psychopathologies[69,73,74]. Indeed, the cerebellum is involved in cognition, multimodal sensory and sensory motor integration, and other functions that are affected in neuropsychiatric

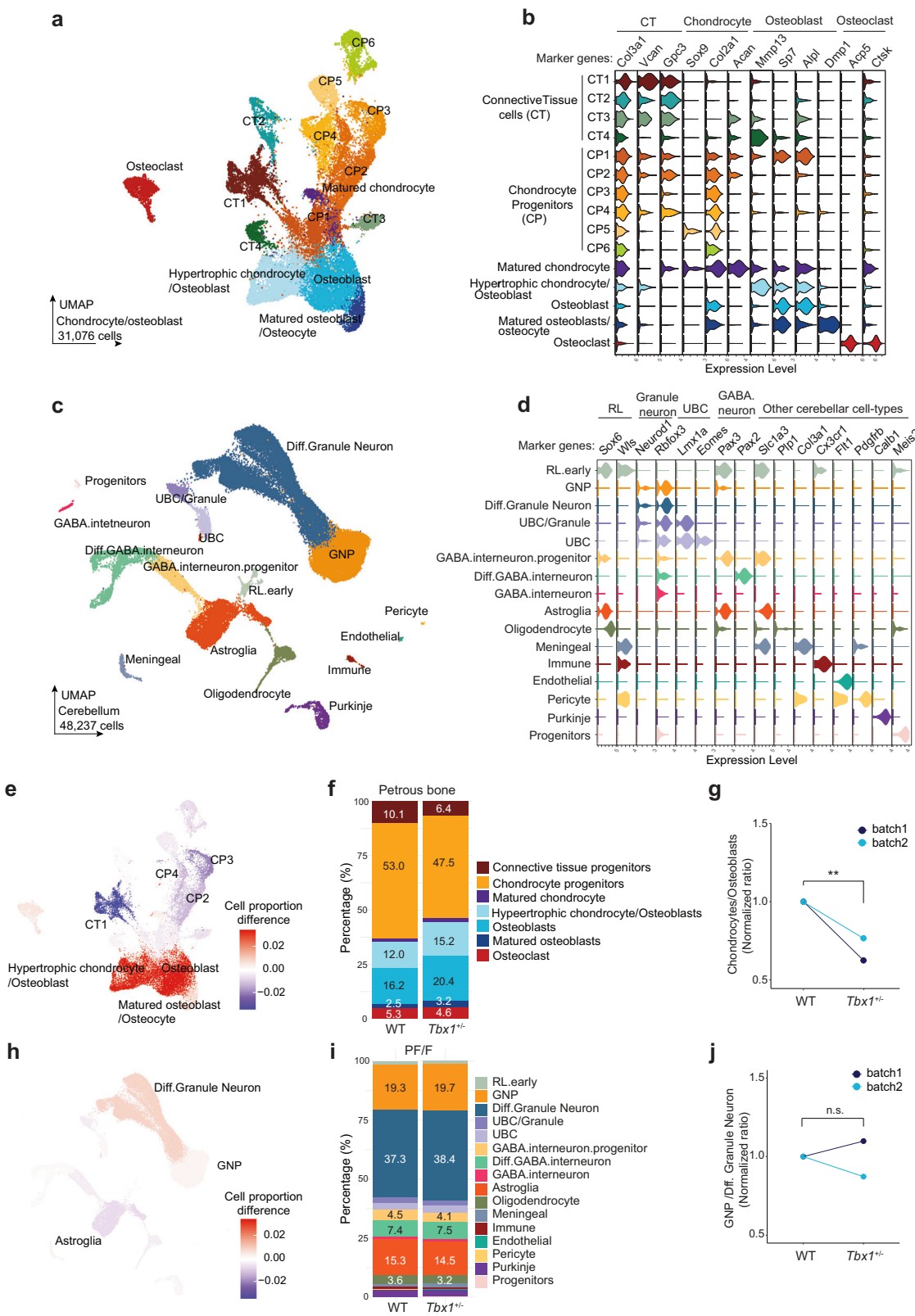

disease[35,36]. It also plays a major role in motor learning, and its function extends to cognitive and emotional controls[75-77]. Although cerebellar dysfunction in 22q11DS and SCZ is heavily implicated, the specific cerebellar regions, their crosstalk among tissues, or circuits that underlie the symptoms are unknown. Here we report a local and robust structural defect (~70% reduction) of the vestibular cerebellum lobules PF/F and an associated deficit in cerebellar-dependent motor learning

in 22q11DS mice. Despite this significant dysplasia, no abnormalities in cell composition and neurogenesis have been detected in the PF/F of 22q11DS models. However, we found a deformity of the surrounding skeletal structures, namely the SF of the PTB, which encapsulates the PF/F and the SCs in the periotic capsule. We also show that haploinsufficiency of the 22q11DS gene *Tbx1* recapitulated the structural and motor-learning deficits in 22q11DS mice.

**Fig. 7 | *Tbx1* haploinsufficiency leads to the cell-lineage shift of chondrocytes to osteoblasts in the petrous temporal bone but does not affect cell-type composition in the paraflocculus/flocculus.** a–j Comparative snRNA-seq analysis using the petrous temporal bone (PTB) (**a**, **b**, **e**–**g**) and PF/F (**c**, **d**, **h**–**j**) from P5.5 wild-type (WT; *n* = 10 mice from two batches) and *Tbx1*[+/−] mice (*n* = 10 mice from two batches). **a** UMAP visualization of chondrocytes and osteoblasts from PTB tissues, comprising 31,076 cells (WT and *Tbx1*[−/−] cells combined). Visualization segregates cells into annotated cell types differentiated by color coding, including four connective tissue (CT) subpopulations, six chondrocyte progenitor (CP) subpopulations, matured chondrocytes, osteoblasts, and osteoclasts. **b** Violin plots show marker gene expression across cell types (as in **a**). **c** UMAP visualization of cerebellar cells from PF/F tissues, comprising 48,237 cells (WT and *Tbx1*[+/−] cells combined). Visualization segregates cells into annotated cell types distinguished by color coding, including granule neuron progenitors (GNP) and differentiated (Diff.) granule neurons, GABAergic interneurons and their progenitors, unipolar brush cells (UBC), rhombic lip progenitors (RL), etc. **d** Violin plots show marker gene expression across cell types (as in **c**). **e** UMAP visualization of the changes in chondrocytes and osteoblasts in the PTB of *Tbx1*[+/−] mice, compared to that of WT mice. **f** Bar chart illustrates the proportions of cell types between WT and *Tbx1*[+/−] cells in the PTB, showing decreased chondrocyte populations and increased osteoblast populations. **g** Normalized ratio of chondrocyte/osteoblast populations of WT and *Tbx1*[+/−] cells in the PTB, showing a marked decrease in this ratio in *Tbx1*[+/−] mice across two batches. Each batch is pulled from five mice per group. DCATS (beta-binomial regression-based test); **FDR = 0.008. **h** UMAP visualization of the changes of cerebellar cell populations in the PF/F of *Tbx1*[+/−] mice, compared to that of WT mice. **i** Bar chart illustrates the proportions of cell types between WT and *Tbx1*[+/−] cells in the PF/F. **j** Normalized ratio of GNPs/Diff. granule neurons of WT and *Tbx1*[+/−] cells in the PF/F, showing no significant changes between the groups. DCATS (beta-binomial regression-based test); FDR = 0.85. FDR false discovery rate, n.s. not significant. Source data are provided as a Source Data file.

The cerebellar flocculus (F) is a pivotal structure in the central networks that modulate cerebellar-dependent, error-driven motor learning and the oculomotor systems that stabilize images on the retina during head movements including VOR[78–81]. A possible role of the paraflocculus (PF), especially the ventral PF, as a part of the PF/F functional complex, is also implied in the regulation of VOR[82–84]. In response to image instability (retinal slip), the VOR undergoes adaptive changes in gain (amplitude) to restore distortion-free performance. Visual motion in the opposite direction of head motion results in a learned increase in VOR gain to cancel retinal slip, whereas visual motion in the same direction of head motion results in a learned decrease in VOR gain to cancel retinal slip. Here we show that the volumetric reduction of the PF/F in 22q11DS mice is associated with a deficit in VOR adaptation when the head rotates in the opposite but not the same direction as visual motion. Interestingly, the VOR adaptation deficit in 22q11DS mice was not due to dysfunction in vestibular-induced oculomotor control, as their baseline VOR performance was normal. These results suggest that 22q11DS affects the plasticity mechanisms within the cerebellar circuits.

Because lesions in the PF/F impair smooth pursuit (i.e., gaze-holding that generates gaze-evoked nystagmus) and VOR[78,84–87] in several species, our data imply that PF/F dysplasia leads to the VOR adaptation deficit in 22q11DS, most likely through synaptic plasticity defects within the PF/F circuitry. This mechanism could be relevant to SCZ as 22q11DS substantially increases the risk of SCZ[6,9], and impaired smooth visual pursuit of a moving target is associated with psychosis[88–94].

Multiple plasticity mechanisms contribute to cerebellum-dependent motor learning[47]. Long-term synaptic plasticity is implicated in the mechanisms of lasting VOR gain adaptation[95–97]. Previous studies have suggested that VOR gain-increase but not VOR gain-decrease depends on LTD at Pf−PC excitatory synapses[46,97,98]. In support of this notion, LTD but not LTP in *Df(16)1/+* and *Tbx1*[+/−] mice was deficient. However, given that *Tbx1* is either not expressed or expressed at low levels in the adult cerebellum (see below), it is unclear exactly how *Tbx1* haploinsufficiency causes cerebellar LTD defects. The disparate, context-dependent VOR learning deficits in 22q11DS most likely arise from the pathologic alteration of some, but not all, forms of Pf−PC synaptic plasticity. Differences in the development or maintenance of synaptic inputs to PCs could also contribute to VOR learning deficits. However, we observed functionally intact Pf- and Cf-synaptic responses in PCs of *Df(16)1/+* and *Tbx1*[+/−] mice, suggesting that these synaptic inputs are grossly normal. Nonetheless, because plasticity at other synapses originating in multiple cerebellar regions could be involved, detailed circuitry and synaptic plasticity mechanisms in 22q11DS remain to be elucidated.

Although TBX1, a transcription factor in the T-box family, is either not expressed or expressed at very low levels in adult neurons, it has an essential role in early vertebrate development[66]. During early embryonic stages, TBX1 is present in the pharyngeal endoderm, mesodermal core of pharyngeal arches, head mesenchyme, secondary heart field, sclerotome, and otocyst[99–102]. In mice, *Tbx1* expression was detected in the endothelial cells of the brain vasculature[103]. In the developing brain, *Tbx1* was undetectable during early embryonic stages but robustly expressed at P25 in the forebrain[103,104] and was not found in the developing hindbrain[105]. Our snRNA-seq data have not detected *Tbx1* expression in the PF/F and cerebellum, except the low levels in the rhombic lip at E10, further suggesting a limited presence of *Tbx1* in the cerebellum. *Tbx1* copy number variations (global loss or gain of function) lead to the imbalance of cell proliferation and differentiation in anterior/secondary heart field[102,106,107], pharyngeal endoderm[102], and inner ear[102,108]. *Tbx1* deficiency results in many physical manifestations of 22q11DS, including cardiac outflow tract anomalies[101,102]; pharyngeal apparatus defects[109,110]; craniofacial and skeletal defects[111–113]; hypocalcemia[111]; hypoparathyroidism[114]; abnormalities in the development of the outer, middle, and inner ear[56,100,108,115–118], and other anomalies[114,119], which makes *Tbx1* a key genetic determinant of 22q11DS[26,114,120,121]. Despite its virtual absence in adult neurons[122] *Tbx1* deficiency also affects cognitive function, such as sensorimotor gating in the form of deficient pre-pulse inhibition of the acoustic startle reflex[103], and other autism-spectrum disorder− or SCZ-associated behaviors[123–126] in mice. In humans without 22q11DS, rare variants of *Tbx1* have been associated with autism-spectrum disorder[127,128] and SCZ[129].

In 22q11DS mice, we found a substantially shallower SF within the temporal bone and a subtle but significant reduction of the SCs in the inner ear. This is consistent with previous findings of *Tbx1*-dependent skeletal and inner ear defects[56,100,108,111–113,115–118]. It also suggests that the cerebellar PF/F, temporal bone, SF, and SCs are dynamically inter-linked during embryonic and early postnatal development, and this relation depends on the *Tbx1* dosage.

The spatial and functional relation among the SF, SCs, and PF is well-documented. In mice, the SF extends laterally through the arc of the ASC, and the floor and posteromedial wall of the SF are formed by the otic capsule surrounding the LSC and PSC in the inner ear[51]. The SF is lined with dura mater and houses the petrosal lobule of the PF[34,50]. In rodents, the SF develops before and independently of the development of the PF, as the PF occupies the already-formed pocket area within the SF[50]. The cavity filled with loose, vascularized connective tissue is called the primitive SF at E16.5; by E18.5, the definitive SF is formed. The SF and PF have no contact until the PF invades the SF cavity at P3.5. Our snRNA-seq data have shown that *Tbx1* expression was detected in the postnatal PTB but not in the postnatal PF. *Tbx1* was detected only at an early embryonic stage (E10) in the rhombic lip, a unique progenitor zone that gives rise to excitatory neurons in the cerebellum[130] Given that only the PF/F but no other cerebellar lobules (Crus I, Crus II, Vermis) is affected in 22q11DS and *Tbx1*[+/−] mice, we deduce that *Tbx1* haploinsufficiency may not directly affect PF

development. It may instead disturb its surroundings, for instance, by making the SF shallower. Thus, the most parsimonious explanation of the PF/F dysplasia in 22q11DS is that the shallower SF pocket limits the space for the pedunculated growth of the PF and thus stunts its development. Indeed, we found a strong correlation between the SF and PF/F in mice, suggesting their interrelated structural development. Alternatively, during development, the pedunculated growth of the PF may expand the surrounding SF and SCs, particularly the ASC, as the PF invades the SF and displaces the connective tissue and dura mater therein. In this scenario, the SF would expand to accommodate PF growth.

Compared to the PF/F dysplasia in 22q11DS mice, that in humans with 22q11DS was less profound. The interspecies differences may be explained by the relatively high level of functioning (IQ >70) in humans with 22q11DS, which may simply reflect the inclusion of patients with more developed cerebella. Alternatively, differences could be due to disparate anatomy or development of the PF/F. In mice, the PF/F forms as bulb-shaped protrusions in the ventrolateral part of the cerebellum; this mainly occurs at P0 to P6, but the PF/F further expands laterally during cerebellar development[34,131]. In humans, the F includes the accessory PF (also called the ventral PF) and the tonsil, which correspond to the dorsal PF in nonhuman mammals and are located in the cerebellopontine angle[132].

The PF–SF–SC relation also differs between mice and humans. In humans, canal growth outpaces fossa growth because the SF and petrosal lobule are absent in adults. However, a fossa resembling the murine SF is present during the fetal and neonatal stages of human development. This fossa is subsequently filled with a bone to form the petromastoid canal[133]. Given the differential regulation and development of these skeletal and cerebellar structures between rodents and humans, we suggest that the relation among the PF, SF, and SC during development is weaker or more transient in humans than it is in mice, and this may explain the smaller degree of the PF/F dysplasia in humans with 22q11DS compared to mouse models of 22q11DS.

Despite the robust PF/F dysplasia associated with *Tbx1* haploinsufficiency in mice, we detected no defects in neural cell composition, proliferation, apoptosis, migration, or differentiation in the PF/F. However, we identified a significant cell-lineage shift from chondrocytes to osteoblasts in the PTB, suggesting that *Tbx1* haploinsufficiency affects cell-fate determination by leading to precocious differentiation of chondrocytes to osteoblasts[134,135]. The petrous part of the temporal bone houses the bony labyrinth of the inner ear, which is formed by multiple layers of developmental cellular processes: the otic vesicles form the membranous labyrinth (cartilage) that ossifies to the bony labyrinth (bone), through the periotic mesenchymal cell-mediated endochondral ossification (cartilage template to bone formation). *Tbx1* is expressed in the otic vesicles and periotic mesenchyme, and *Tbx1* depletion in mice is associated with abnormalities in the inner ear and temporal bone development[56,113,114,136]. For instance, *Tbx1*-dependent chondrocyte maturation in the spheno-occipital synchondrosis during the development of the cranial base is involved in chondrocyte hypertrophy and osteogenesis[136].

A notable caveat of our single-cell methodologies is the inherent challenge of detecting subtle changes in gene expression between conditions. This limitation is particularly pertinent in the context of *Tbx1* haploinsufficiency, where the differential gene expression may not always be pronounced. Hence, although our study provides a comprehensive snapshot of the cellular landscape alterations, the resolution of gene expression changes warrants cautious interpretation. In addition to snRNA-seq studied here, other omics approaches, like assay for transposase-accessible chromatin with sequencing at the single-cell level in the PTB, may further delineate cell status and dynamics between chondrocytes and osteoblasts and provide substantial insights into the regulatory gene network.

Several previous studies used a conditional deletion or over-expression approach to show that *Tbx1* is important for progenitor maturation and fate determination during early embryonic development of the heart, head mesenchyme, skeleton, otic vesicle, ear, or vasculature[56,102,108,122]. However, these effects were observed only in mice with conditional knockout of both copies of *Tbx1*; no defect was found if only one copy of *Tbx1* was conditionally deleted. Similarly, conditional hemizygous *Tbx1* deletions did not affect several cellular lineages in the PF/F morphology. The genetic approach to identifying a cell-specific role of *Tbx1* and to mimic and rescue the phenotypes by using the Cre-lox system in vivo has failed in our hands, which could be due to several limitations that hinder the interpretation of those results. Manipulation of *Tbx1* expression in certain Cre lines (*Mesp1^Cre*, *Twist2^Cre*, and *Foxg1^Cre*) leads to lethality. Furthermore, the negative results may be caused by incomplete gene excision of the floxed loci or inappropriate timing of recombination. *Tbx1* manipulation in these lineages also may not specifically affect chondrocyte-to-osteoblast differentiation. Thus, more refined tools or approaches to control *Tbx1* dosage in a spatiotemporal manner in vivo will be needed to help understand the roles of *Tbx1* during cerebellar development.

In conclusion, we show that haploinsufficiency of the 22q11DS gene *Tbx1* impairs the normal development of the cerebellar PF/F and the part of the temporal bone that houses the PF/F. *Tbx1* haploinsufficiency also leads to deficits in long-term synaptic plasticity in PF/F and cerebellar-dependent motor learning. Thus, our study demonstrates disruption of locally interrelated skull and brain development and their structural and functional deficiencies in 22q11DS. We propose an important bone/brain relation in which a cranial malformation may lead to functional and behavioral deficiencies by stunting cerebellar lobule development.

## Methods

### Ethical considerations

The care, handling, welfare monitoring, methods of euthanasia, and all procedures involving animals were approved by the St. Jude Children's Research Hospital (protocol #3177) or Max Planck Florida Institute for Neuroscience (protocol #15-205) Institutional Animal Care and Use Committee and adhered to principles described in the National Institutes of Health Guide for the Care and Use of Laboratory Animals. All ethical regulations relevant to human research participants were followed. The human subject portion of the study was approved by the Institutional Review Board of the University of Pennsylvania (Penn) (protocol #812481).

### Animals

Both male and female mice (aged P3–8 months) were used for all experiments, as there were no sex-specific effects on PF/F volumes in the 22q11DS and *Tbx1^+/−* mutants (Supplementary Figs. 1a, 3d). The generation of *Del(3.0 Mb)/+* [38], *LgDel/+* [26], *Df(16)1/+* [25], *Znf74l-Ctp/+* [137], *Rtn4r^+/−* [138], *Comt^+/−* [139], *Gnb1l^+/−* [103], *Df(16)2/+*, *Df(16)3/+*, *Df(16)4/+*, *Df(16)5/+*, *Dgcr8^+/−*, *Tbx1^+/−*, *Sept5^+/−* [121,140], *Tbx1^fl/+* [56], *Tbx1-GFP^fl/+* [63], *Mesp1^Cre* [61], *Tie2^Cre* [57], *NeuroD1^Cre* [141], *L7^Cre* [58], *Twist2^Cre* [142], *Lyz2^Cre* [59], *Foxg1^Cre* [62], and *p53^+/−* [143] mouse lines has been reported previously. *All* mouse strains (except *LgDel/+* mice) *were backcrossed* onto the C57BL/6J genetic background for several generations. *LgDel/+* mice were *maintained* on the C57BL/6N genetic background for several generations. Mutated alleles were transmitted paternally or maternally, and no difference was found between transmissions. The *LgDel* mutation was transmitted paternally. Mice were group housed by sex with ad libitum access to food and water on a 12-h light/dark cycle at 21–24 °C and a humidity of 40–45%.

### Generation of *Arvcf-Txnrd2/+* and *T10^+/−* mice

The *mArvcf-Comt-Txnrd2*−deletion mouse model was generated using CRISPR-Cas9 technology and direct zygote injection. Briefly, a mixture

of the single-guide RNAs (sgRNAs) and Cas9 protein consisting of 60 ng/mL Cas9 protein (St. Jude Protein Production Core) and 20 ng/mL of each sgRNA was injected into C57BL/6J zygotes. Animals were genotyped by PCR and gel electrophoresis for the deletion-specific amplicon. Animals positive for the desired deletion were confirmed by targeted deep sequencing, backcrossed to C57BL/6J mice, and then bred to homozygosity. The following sequences are the sgRNA spacers and associated genotyping primers used: CAGE114.Arvcf.g55 spacer (5′-GAGAAGACCTGAGTACGGGC-3′), CAGE126.Txnrd2.g6 spacer (5′-GATCTCTTGGTGATCGGTGG-3′), CAGE114.Arvcf.F (5′-GCGGCCGCTC TCTGCCTGAG-3′), CAGE126.Txnrd2.R (5′-TGTTCCTGGAACATTCTGC TCCTGG-3′).

The *T10* (*Tango2*)-knockout mouse model was engineered using CRISPR-Cas9 technology and delivery into pronuclear-stage zygotes. Two sgRNAs (T10_5End_108 and T10_3End_179; 50 ng/μL) and Cas9 mRNA transcripts (100 ng/μL) were delivered in pronuclear-stage C57BL/6J zygotes via microinjection, as described previously[144]. Mice obtained from these microinjections were characterized at the genomic level using PCR amplification and Sanger sequencing. Mice positive for the desired deletion were subsequently bred to C57BL/6J mice for a few generations and mated to generate homozygous mice. Guide sequences used to target T10 are as follows: T10_5End_108, TTCTAGTGCCAGTTACACGC; T10_3End_179, TTACACATGAGAACC TACTC. Genotyping primers used to amplify the Intron 2 cut site are as follows: T10_Cas9_F1, GCAGGCTGCTGGGTGTGTGTCCTCTGA; and T10_Cas9_R2, TCCTGATAACAACCCCAGCTCTCTGGA. Genotyping primers used to amplify the Intron 8 cut site are as follows: T10_Cas9_F2, GCTTCTGGCTGCCTCACCTGTTCA; and T10_Cas9_R1, ATACCAATTTACCACATTCCTTACA. The T10 deletion was detected using primers T10_Cas9_F1 and T10_Cas9_R1, generating a PCR amplicon of ~650 bp.

## Rodent magnetic resonance imaging analysis

The animal MRI study was performed using a 7T Bruker ClinScan system (Bruker BioSpin MRI GmbH, Ettlingen, Germany) equipped with a 12S gradient coil. A mouse brain surface coil (Bruker BioSpin) was used. Animals were anesthetized and maintained with 1.5–2% isoflurane during the experiments. Transverse T2-weighted turbo spin-echo images were acquired for volume measurements (Repetition time/Echo time = 3841/50 ms, field of view = 25 × 25 mm, matrix = 320 × 320 pixels, echo train length = 7, number of slices = 42, number of averages = 1, thickness = 0.4 mm, scan time = 6.5 min).

For ex vivo high-resolution MRI studies, mice were deeply anesthetized by administrating Avertin [tribromethanol/amylene hydrate, 1.25% solution, 0.2 mL/10 g bodyweight, intraperitoneally (i.p.)] and transcardially perfused with 4% paraformaldehyde (PFA) in 0.1 mol/L phosphate buffer(PB, pH 7.4). After decapitation, a mouse head was post-fixed in 4% PFA overnight and transferred to PB (pH 7.4). The fixed mouse head was placed in a 15-mL tube with fluorinert (3 M, Maplewood, MN). Using a 7T Bruker ClinScan system (Bruker BioSpin) equipped with a 12S gradient coil and a mouse head volume coil (Bruker BioSpin), T2-weighted turbo spin-echo 3D images were acquired overnight for volume measurements (TR/TE = 1800/70 ms, FOV = 15 × 22, matrix = 252 × 384, NEX = 6, scan time = 16 h). The regional brain volumes (Crus I, Crus II, Vermis IV/V, and PF/F) were obtained by manually segmenting each region and computing the volumes using OsiriX (v5.7, Pixmeo, Bernex, Switzerland) and 3D slicer[145]. All volumes were measured using manual segmentation of the MRI data, except the data from the machine learning–based measurement in Supplementary Fig. 1g, h.

## Human magnetic resonance imaging analysis

All human subjects underwent informed consent prior to their participation in the study. They received modest financial compensation

($100 gift card) for their participation. Structural correlates in the human population were investigated using neuroimaging data from the prospective study Brain-Behavior and Genetic Studies of the 22q11DS performed at the Penn and the Children's Hospital of Philadelphia (CHOP). This study has been described previously[70]. Briefly, inclusion criteria were age >8 years, intelligence quotient (IQ) >70, and medical stability at the time of enrollment; subjects were excluded if they had IQ <70 or serious neurological disorders (e.g., uncontrolled seizures, head trauma, CNS infection, or intracranial neoplasm). A total of 80 subjects with 22q11DS were included (mean age 22.2 years, SD 8.7 years, 38 males). Deletion status was confirmed using multiplex ligation-dependent probe amplification[146]. For comparison, a group of nondeleted, typically developing (TD) control subjects was obtained from the Philadelphia Neurodevelopmental Cohort, a large prospective study that acquired MRIs by using the same equipment and structural imaging protocol[147]. A total of 68 TD subjects with high-resolution MRIs were included (mean age 18.9 years, SD 3.5 years, 42 males). Study procedures for both 22q11DS and control samples were approved by the institutional review boards at both Penn and CHOP.

High-resolution MRIs were acquired from both groups by using the same 3T MRI scanner (TIM Trio; Siemens, Erlangen, Germany) and 32-channel head coil. T1-weighted magnetization prepared rapid acquisition gradient echo (MPRAGE) sequences were acquired with the following parameters: TR/TE 1810/3.51 ms; inversion time 1100 ms; FOV (field of view) 180 × 240 mm; effective resolution 1 mm$^3$. Images were converted to NIFTI files and imported into Mango version 4.1 (Lancaster, Martinez; www.ric.uthscsa.edu/mango) to define regions of interest (ROIs) in the PF/F. ROIs were defined by a board-certified neuroradiologist (JES) blinded to diagnosis and with >20 years of experience in quantitative neuroimaging. Left and right PF/Fs were defined separately but subsequently combined into a single measure. ROIs were primarily defined in the sagittal plane, but orthogonal planes were also used for quality assurance; this was particularly useful for differentiating isointense cisternal cranial nerves from adjacent cerebellar parenchyma. Measurements were repeated in a subsample of 25 subjects, with excellent intra-rater reliability (ICC = 0.95; 95% CI [0.89–0.98]).

To measure total cerebellar and lobular volumes, images were imported into the CIVET pipeline (v2-1-1) for automated skull stripping and nonuniformity correction[148–150]. After visual inspection, cerebellar segmentation was subsequently performed using MAGeT[151,152], an automated, multiatlas cerebellar parcellation algorithm. MAGeT estimates cerebellar volumes based on five high-resolution MRI atlases (three female, two male), with each atlas defined by two trained anatomists blinded to each other's work. The algorithm first transforms each atlas to a subset of the sample (21 randomly selected subjects) via nonlinear registration to generate a template library (5 × 21 = 105 templates). This step is advantageous in that it helps account for individual variation in cerebellar morphology and subsequently improves image registration. A second nonlinear registration step was then performed to propagate ROIs between templates and individual subjects. Measures of total cerebellar volume obtained via MAGeT have a very high agreement (k = 0.93) compared to gold-standard manual tracings[152].

Volumetric data were imported into the R statistical environment[153]. Summary statistics were then calculated. Group differences in both PF/F and total cerebellar volumes were compared via multiple linear-regression models, while simultaneously controlling for the effects of age and sex. Relative group differences in PF/F volumes were assessed by incorporating total cerebellar volume as an additional covariate.

As a complementary approach to the hypothesis-driven analysis described above, we performed voxel-based morphometry (VBM) using the spatially unbiased infratentorial template (SUIT) toolbox[154,155]. SUIT functions within the statistical parametric mapping

(SPM12) software package, which in turn relies on MATLAB® (vR2020a; MathWorks, Natick, MA) for some functions. Unlike ROI-based analysis, VBM is data-driven, examining regional group differences in morphology at high spatial resolution and without a priori neuroanatomic constraints[156]. However, VBM has several limitations, particularly its reliance on many complex image-processing steps and the inherent risk of false positives due to the large number of statistical tests performed. SUIT-based VBM ameliorates these issues by improving cerebellar registration and constraining the search space relative to whole-brain VBM[154]. Using standard SUIT image-processing functions, we isolated the cerebellum and segmented its gray matter. After visual inspection, images were normalized to the SUIT cerebellar template and resliced into the atlas space. Images were then smoothed in SPM using an 8-mm kernel. To account for global effects on individual voxels, proportionate-scaled mean gray matter density was included as a global covariate. Age and sex also were included as covariates. Multiple testing was controlled with a family-wise error threshold of $p = 0.001$ and a cluster (extent) threshold of $k = 100$ voxels. All statistical tests for both ROI-based and VBM were two-tailed.

### Computerized tomography measurement in mice

Mouse skulls were scanned at the Center for In Vivo Imaging and Therapeutics at St. Jude Children's Research Hospital on an Inveon PET/CT system (Siemens), at 14-μm resolution for the high-resolution scans and 45-μm resolution for the low-resolution scans. Alternatively, mice were scanned on a Bruker Skyscan 1276 CT (Bruker Biospin) at 18.59-μm resolution, at 70 kV and 200 μA and using a 0.5-mm aluminum filter. Before scanning, the mice were sedated with 2–3% isoflurane (3% induction, 2% maintenance) and then placed on an integrated animal-handling system with a camera for monitoring.

Using the 3D analysis program within the Inveon Research Workplace software (IRW 4.2, Siemens) and 3D Slicer (Surgical Planning Laboratory, Boston, MA), the volumes of SF spaces and the height and width of individual SCs (anterior, posterior, and lateral) were measured by drawing ROIs in the 3D volumes. Evaluations of the PTB and skull volumes were done in the analysis program within IRW or 3D Slicer (Surgical Planning Laboratory). PTB volumes were measured by drawing an ROI around the PTB and setting a minimum threshold of 800 HU to segment the skull, and then removing parts of the skull segmentation that were not PTB. Total-skull volume was determined by drawing an ellipsoid-shaped ROI over the head and setting a minimum threshold of 800 HU. The total volume of the >800 HU region was used as the total-skull volume.

### 3D rendering of mouse brain and skull

The parcellation of mouse brain structures was performed using DeepBrainIPP[37]. Briefly, we used a deep-learning (U-Net–based) method for skull stripping (i.e., digitally separating the brain from the MRI that includes both the skull and brain). The extracted brain was then subjected to image registration to an atlas by using Advanced Normalization Tools[157]. Image registration provides comprehensive brain structure masks. The implementation details and code are available at https://github.com/stjude/DeepBrainIPP. A marching cubes algorithm was used to generate 3D-rendered surfaces (as displayed in Fig. 1b and Supplementary Movies 1–4) by using Matlab. For 3D visualization of the mouse brain with skull, the CT images were cropped, and a threshold was set in ImageJ 1.54f[158] before fiducial registration of the CT and MRI data in Slicer 5.2.2[145]. Registered images were then combined into separate channels of a TIFF file in ImageJ before loading into Imaris 10.0.1 (Oxford Instruments) for movie generation.

### Video-oculography, vestibulo-ocular reflex training, and gain adaptation

Video-oculography and the VOR study were adapted as previously described[159]. Briefly, for behavioral experiments, stainless steel head-posts were attached to the top of the skull of anesthetized animals during surgery; postoperative analgesia was provided by injection of carprofen and buprenorphine (SR LAB, Santa Barbara, CA). After a 2-week recovery period, mice were head-restrained using their surgically implanted head-posts on a custom-made VOR apparatus. The apparatus included a motorized rotation stage (T-RSW60C, Zaber Technology, Vancouver, BC) to deliver horizontal vestibular stimuli and a machine-vision camera directed to the left eye so that its position could be tracked in response to passive head turns. Eye movement was determined by pupil position, which was computed by eye-tracking software (ETL-200; ISCAN Inc, Burlington, MA); the angular position of the eye was derived using a previously published method that accounted for eye curvature[160]. An infrared LED was fixed to the top of the camera to provide illumination, and a reference corneal reflection was used for analysis procedures. Prior to experiments, pilocarpine (2% ophthalmic drops; Patterson Veterinary Supply, Loveland, CO) was briefly applied (<1 min) onto the eye to limit pupil dilatation, so that the eye could be accurately tracked in darkness. For visual-vestibular training, custom-written software generated a high-contrast grating, consisting of black and white vertical stripes, that was presented on monitors placed in front of the animal. The grating was moved relative to the stage position, either out-of-phase with the vestibular stimulus or in-phase with the vestibular stimulus. Immediately prior to training, a test measurement at 1-Hz frequency was obtained to gauge baseline VOR performance. After training (60 min of 1-Hz frequency, 1.5× visual and vestibular stimuli difference for the opposite-direction mismatch training; 60 min of 1-Hz frequency, 0× visual and vestibular stimuli difference for the same-direction mismatch training), the VOR was re-tested in darkness. Changes in VOR amplitude were computed as the percentage difference in VOR performance after training, relative to the baseline measurement before training (ΔVOR). This normalization procedure facilitated comparisons across sessions and between mice. For each mouse, the order of training sessions was randomized, with at least 2 days between tests for all conditions.

### Patch clamp electrophysiology

**Cerebellar slices.** Mice were rapidly decapitated, and brains were removed and placed in cold (4 °C) dissecting media containing (in mM) 125 choline-Cl, 2.5 KCl, 0.4 CaCl₂, 6 MgCl₂, 1.25 NaH₂PO₄, 26 NaHCO₃, and 10 glucose (300–310 mOsm), equilibrated with 95% O₂/5% CO₂. Coronal cerebellar slices (400 μm) were made using a vibrating microtome (Leica VT1200, Teaneck, NJ). Slices were transferred to artificial cerebrospinal fluid (ACSF) containing (in mM) 125 NaCl, 2.5 KCl, 2 CaCl₂, 2 MgCl₂, 1.25 NaH₂PO₄, 26 NaHCO₃, 10 glucose (300–310 mOsm), equilibrated with 95% O₂/5% CO₂ at 34 °C for 30 min, followed by 1 h at room temperature prior to use. Slices were transferred to a recording chamber mounted on a BX51WI upright microscope (Olympus Life Science, Center Valley, PA) and superfused (1–2 mL/min) with warm (30–32 °C) ACSF. Slices were viewed with a CCD (charged-couple device) camera (Rolera-XR, QImaging, Burnaby, BC) using IR-DIC (infrared–differential interference contrast) optics. Purkinje neurons in the PF were identified by soma shape, size, and location in the Purkinje cell (PC) layer. Synaptic responses were evoked by current pulses (intensity 0.1–1 mA, duration, 100 μs) delivered to glass electrodes filled with ACSF by using a stimulus isolator (Iso-flex; A.M.P.I., San Francisco, CA). Parallel fiber (Pf) inputs were stimulated by a glass electrode placed in the molecular layer, and climbing fiber (Cf) inputs were stimulated by a glass electrode placed in the granule cell layer near the recorded PC. Pf stimulation of sufficient strength resulted in a simple postsynaptic spike, and Cf stimulation resulted in a complex postsynaptic spike.

**Whole-cell recording.** Whole-cell current-clamp recordings were made with patch pipettes (3–5 MΩ) by using a Multiclamp 700B

amplifier, digitized (10 kHz) with a Digidata 1440, and recorded using pCLAMP 10 software (all Molecular Devices, San Jose, CA). The internal solution contained (in mM) 115 potassium gluconate, 20 KCl, 10 HEPES, 4 $MgCl_2$, 4 ATP-$Mg_2$, 0.4 GTP-Na, and 10 creatine phosphate-$Na_2$ (pH 7.4, 290–295 mOsm). Membrane potentials were corrected for a liquid junction potential of −10 mV. Pipette capacitance and series resistance were compensated using the amplifier's circuits. Input resistance, series resistance, and membrane potential were monitored during recordings.

**Synaptic plasticity.** Cells were held at −80 mV by current injection to limit action potential firing during baseline and after plasticity induction. Baseline Pf responses were recorded at 0.067 Hz for 5–10 min, and stimulus strength was adjusted such that Pf responses were similar for each recording. Cf stimulus strength was adjusted to ensure a complex spike was elicited. LTD was induced by conjunctive stimulation of Pf and Cf inputs. A PF tetanus (6 stimuli, 100 Hz) was followed by a single Cf stimulus (120 ms after the third PF stimulus), repeated 300 times at 1 Hz. LTP was induced by the same Pf-tetanus protocol without Cf stimulation. Responses were quantified by measuring the initial slope of the Pf-induced EPSP. For each cell, responses were normalized to baseline and averaged for each minute of recording. Plasticity was quantified as the EPSP slope 30–40 min after LTP or LTD induction, as a fraction of the baseline slope.

**Histologic analysis and immunohistochemistry**
Mice were deeply anesthetized with Avertin, intracardially perfused with 4% PFA in 0.1 mol/L PB (pH 7.4). The mice were decapitated, and the brains were fixed overnight. Paraffin sections, including the PF/F (7 μm), were deparaffinized and subjected to heat-mediated antigen retrieval in sodium citrate buffer (10 mM, pH 6.0) at 80 °C for 20 min, cooled to room temperature, and washed in 1× PBS for 20 min. Sections were incubated in PBS-blocking buffer (5% goat serum, 3% bovine serum albumin [BSA], 0.2% Triton X-100, in PBS) for 1 h at room temperature and incubated with the following primary antibodies: cleaved caspase-3 (Cell Signaling, Danvers, MA; 9661S, 1:250), PH3 C-2 (Santa Cruz Biotechnology, Dallas, TX; sc-374669, 1:100), Ki67 (Abcam, Waltham, MA; ab15580, 1:500), calbindin D-28K (Millipore, Burlington, MA; ABN2192,1:1000), Pax6 (Developmental Studies Hybridoma Bank, Iowa City, IA; AB528427, 1:5), GFAP (DAKO, Carpenteria, CA; Z0334, 1:250), Tbr2 (Abcam, ab23345, 1:500), and beta-III-tubulin (Sigma, St. Louis, MO; T8660, 1:500). Appropriate Alexa dye–conjugated secondary antibodies (Thermo Fisher Scientific, Waltham, MA; 1:1000) were used to detect primary antibody binding. Hematoxylin and eosin staining was performed in accordance with the manufacturer's instructions (Vector Laboratories, Burlingame, CA). DAPI (Invitrogen, Carlsbad, CA) was used as the nuclear counterstain.

**Ex vivo cerebellar granule neuron migration**
The study of cerebellar granule neurons' germinal zone exit and migration was adapted as previously described[161]. Briefly, mice (P7) were rapidly decapitated and cerebella were dissected and soaked in a suspension of 1–3 μg/μL H2B-mCherry DNA construct in Hank's buffered salt solution (HBSS) and electroporated in a platinum-block petri-dish electrode (CUY520-P5; Protech International, Cornelius, NC) by using a square-wave electroporator (CUY21-EDIT; Protech International) with the following program: 5 pulses, 90 V, 50-ms pulse, 500-ms interval. The electroporated cerebella were embedded in 4% low-melting-point agarose in HBSS, and 300-μm coronal sections of the cerebella were prepared on a vibratome (VT1200; Leica microsystems, Wetzler, Germany). The sections, including the PF, were transferred to 0.4-μm Millicell cell culture inserts (Millipore) and incubated in Basal Medium Eagle solution supplemented with 0.5% glucose, 2 mM L-glutamine, 50 U/mL penicillin–streptomycin, 1× B27, and 1× N2 supplements (Thermo Fisher Scientific). After 48 h, the slices

were fixed with 4% paraformaldehyde and mounted on glass slides with Paramount mounting solution (DAKO). Images and the migration distance of H2B-mCherry–labeled CGNs from the pial surface in the PF were acquired by LSM780 confocal microscope (ZeissAxioObserver) and Zen2012 software (Zeiss, Dublin, CA).

**Single-nucleus RNA sequencing**
**Isolation of nuclei from the petrous temporal bone and paraflocculus/flocculus.** P5.5 mice were rapidly decapitated, and the PTB and PF/F were isolated and placed in cold (4 °C) dissecting media containing (in mM) 125 choline-Cl, 2.5 KCl, 0.4 $CaCl_2$, 6 $MgCl_2$, 1.25 $NaH_2PO_4$, 26 $NaHCO_3$, and 10 glucose (300–310 mOsm), equilibrated with 95% $O_2$/5% $CO_2$. After dissection, each sample was flash-frozen and stored at −80 °C until used. The frozen bones (five mice per group) were placed into prechilled tissue tubes (Covaris, Woburn, MA; cat. no. 520001) on dry ice and pulverized using a prechilled CP01 cryoPREP manual dry pulverizer (Covaris, cat. no. 500230). The coarsely powdered bone was placed into a well of a six-well plate with 1 mL CHAPS/salt-Tris solution (ST buffer) containing 980 μL 1% CHAPS (Millipore, cat. no. 220201) mixed with 1 mL 2× ST buffer (292 mM NaCl, 20 mM Tris-HCl [pH 7.5], 2 mM $CaCl_2$, and 42 mM $MgCl_2$), 10 μL 2% BSA (New England Biolabs, Ipswich, MA; cat. no. B9000S), and 10 μL nuclease-free water, and gently cut using Noyes spring scissors (Fine Science Tools, Foster City, CA; cat. no. 15514-12) for 2 min on ice. The nuclei suspension was filtered through a 40-μm strainer into a 15-mL tube (pre-blocked with 2.5% BSA). The filter was washed with 1 mL of 1× ST buffer, and the filtered volume was brought up to 5 mL by adding 1× ST buffer. Following additional filtration through a 30-μm strainer, the solution was centrifuged at 300×g for 5 min. The pellet was resuspended in 1% BSA in PBS (250–500 μL based on pellet size) and filtered through a strainer cap FACS tube (Corning, Corning, NY; cat. no. 352235). Nuclei were counted and checked for integrity under a microscope, and the volume was adjusted to a concentration of $1 \times 10^6$ nuclei/mL. The frozen PF/F (5 mice/group) was placed directly into 1 mL CHAPS/ST buffer (without pulverization), chopped for 3 min on ice, and processed further as described above. All solutions used for nuclei isolation were amended with 0.2 U/μL of RNase inhibitor (Roche, Indianapolis, IN; cat. no. 3335399001). All these processes were repeated using the second set of additional mouse samples for a separate snRNA-seq experiment (a total of 10 mice per group).

**snRNA-seq library preparation and sequencing.** A total of 10,000 nuclei were loaded per channel of a Chromium single-cell 3′ Chip G, and libraries were prepared using a Chromium v3.1 single-cell 3′ kit per the manufacturer's instructions (10× Genomics, Pleasanton, CA). Libraries were sequenced on an Illumina NovaSeq6000 Sequencing System (Illumina, San Diego, CA).

**Processing of 10× genomics snRNA-sequencing data.** We utilized the Cell Ranger software (v7.0.0; 10× Genomics) to process raw sequencing data, which produced demultiplexed FASTQ files. Reads were mapped to the mouse genome (mm10) using Cell Ranger count with intron mode. We then engaged the standard Seurat workflow[162] to process gene count matrices.

To filter out low-quality cells, we retained only cells meeting specific criteria for unique molecular identifiers and a number of detected genes. We excluded nuclei expressing <1000 genes and outliers with a median absolute deviation >3 regarding the total UMI count, the number of genes detected, and the percentage of reads mapped to mitochondrial genes. We also removed genes expressed in <50 cells. To mitigate ambient RNA contamination and have a corrected gene matrix for downstream analysis, we employed SoupX software (v1.6.2)[163]. For dimension reduction and clustering, we used the top 50 principal components as input for Louvain graph-based

clustering, setting the resolution parameter at 1.0. We used Uniform Manifold Approximation and Projection (UMAP) for visualization purposes.

**Cell-type annotation and functional-enrichment analysis.** We initiated cell-type annotation by referencing the single-cell mouse atlas[164] and employing Seurat's TransferData function. Subsequently, manual curation based on cell-type markers per Louvain cluster enabled us to finalize the cell-type annotations. Within each Louvain cluster, we identified differentially expressed genes that were specific to that cluster by using the FindAllMarkers function in Seurat (Wilcox test employed).

We conducted functional enrichment using the clusterProfiler package (v4.6.2)[165] in R, the Gene Ontology biological processes, and hallmark and curated gene sets from MSigDB (v7.4.1)[166,167]. We considered results with a false discovery rate (FDR) <0.05 as significant. The AddModuleScore function in Seurat helped quantify gene sets. To establish a threshold for gene set enrichment, we randomly created 1,000 gene sets that were equal in length to the input gene set and scored each set to derive a permuted null distribution with the enrichment threshold representing the 99th percentile. This threshold was then subtracted from the enrichment scores for visualization, so that all cells >0 represented enriched cells.

**Comparative analysis between Tbx1⁺/⁻ mice and WT mice.** To assess differential cell-type compositions between WT and *Tbx1*⁺/⁻ littermates, we employed DCATS (v1.1.0)[168], a beta-binomial regression-based statistical model specifically designed for single-cell data analysis. DCATS is adept at handling the overdispersion commonly observed in scRNA-seq count data, thereby providing a more accurate representation of cell-type composition differences. For each cell type, DCATS assessed the change in abundance between WT and *Tbx1*⁺/⁻ mice. We set a threshold for significance at FDR <0.1. Our approach to identifying differentially expressed genes between WT and *Tbx1*⁺/⁻ mice conditions involved using Seurat's FindMarkers function for each annotated cell type. We set the thresholds at log-fold change > $\log_2(1.5)$ and *p*.adjust <0.05.

**In vivo drug treatment**
Tranylcypromine (TCP; Sigma P8511) was dissolved in 0.9% w/v NaCl solution and administered (10 mg/kg bodyweight/day) by intraperitoneal injections of pregnant mice at E7.5 to E15.5 or E11.5 to E18.5. One-month-old TCP-treated pups and nontreated controls were examined via MRI to determine their PF/F volumes and via CT to determine their SF volumes.

**Statistical analyses**
We conducted hypothesis testing to compare mean data differences by using Student's *t*-test for two groups and one-way or two-way ANOVA for multiple groups, followed by Holm–Sidak's pairwise comparisons when the normality assumption was met. The Shapiro-Wilk test was employed to assess normality, and Bartlett's test was used to verify homoscedasticity for ANOVA. In cases where assumptions were violated, we resorted to nonparametric alternatives: the Wilcoxon rank-sum test replaced the *t*-test, and the Kruskal–Wallis test was used instead of the one-way ANOVA. For two- or three-way ANOVA, we applied a log transformation to the data prior to its application. Correlations between variables were evaluated using Pearson's correlation coefficient. For assessing differences in proportions between the two groups, we used the Chi-squared test or Fisher's exact test if any expected count was <5. All statistical tests were performed at the standard significance level of 0.05, unless otherwise specified, with FDR corrections applied for multiple testing as necessary. All analyses were conducted using Prism 10 (GraphPad Software, Boston, MA) and R (R Foundation for Statistical Computing, Version 4.3.1), with specific details provided above and/or in the figure legends.

**Reporting summary**
Further information on research design is available in the Nature Portfolio Reporting Summary linked to this article.

## Data availability
All relevant data associated with the published study are present in the paper or the Supplementary Information. The source data underlying the main and Supplementary Figs. are provided as a Source Data file. The RNA-seq data generated in this study are available in the NCBI Gene Expression Omnibus database under accession code GSE254044. The 22q11.2 human subjects' data used in this and previous[70] studies are not publicly available. However, they are included as part of the ENIGMA-22 Consortium dataset, which is available to qualified investigators. Neuroimaging data from the Philadelphia Neurodevelopmental Cohort are publicly available and can be accessed through dbGAP [https://www.ncbi.nlm.nih.gov/projects/gap/cgi-bin/study.cgi?study_id=phs000607.v1.p1]. Additional data relating to this paper are available upon request from the corresponding author, because the size (exceeding 5 TB) of the MRI and CT imaging, immunochemistry, animal behavior, and electrophysiology data are too large to be deposited online. Source data are provided with this paper.

## Code availability
The code used for the analysis of snRNA-seq data were available at Github [https://github.com/ZakharenkoLab/Tbx1_haploinsufficiency_snRNAseq_Project]. Code to perform MAGeT cerebellar volumetric segmentation is public and freely available [https://github.com/CobraLab/MAGeTbrain]. The SUIT toolbox used to perform voxel-based morphometry is similarly publicly available [https://www.diedrichsenlab.org/imaging/suit.htm]. The latter two websites also provide detailed instructions on how to implement these tools on standard MRI datasets.

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

## Acknowledgements

This work was supported, in part, by the National Institutes of Health (R01 MH097742 and R01 DC012833 to S.S.Z.), the Stanford Maternal and Child Health Research Institute Uytengsu-Hamilton 22q11 Neuropsychiatry Research Program grants UH22QEXTFY21 and UH22QEXTFY23 to S.S.Z., a NARSAD Young Investigator Grant from the Brain & Behavior Research Foundation to T.-Y.E., and the American Lebanese Syrian Associated Charities (ALSAC) to S.S.Z. D.R.R. was supported by R01 MH119185. P.A.N. was supported by ALSAC, The Brain Tumor Charity (Quest for Cures), The Mark Foundation (Emerging Leader Award), Alex's Lemonade Stand Foundation (Crazy 8 Initiative), and the National Cancer Institute (P01 CA096832; 1R01 CA270785). C.L. and J.N.S. were supported by ALSAC and the National Institutes of Health grant (P30 CA021765). J.M.C. was supported by the National Institute of Neurological Disorders and Stroke (R01NS123933 and R01NS112289) and the Simons Foundation (SFI-AN-AR-Cross Species-00004551). We acknowledge the Hartwell Center for Biotechnology (St. Jude) for their support with next-generation sequencing. We thank the Zakharenko lab members for constructive comments, Dr. Valerie Stewart for help with mutant mouse production; Drs. David Solecki, Christophe Laumonnerie, Danielle Wong, and Niraj Trivedi for help with the neuronal migration assay; T. Blaine Crowley, Victoria Giunta, Daniel McGinn, and Elaine Zackai for help with patient recruitment and phenotyping; Kosha Ruparel and R. Sean Gallagher for help with human imaging data acquisition and management; John Grime and St. Jude Data Visualization Lab for help with the movies; Briana Williams for help with the graphical abstract; and Dr. Angela McArthur for manuscript editing. The content is solely the responsibility of the authors and does not necessarily represent the official views of the National Institutes of Health or other granting agencies.

## Author contributions

Conceptualization, S.S.Z. and T.-Y.E.; experiments in animals, T.-Y.E., J.S., A.B., S.A., C.M.D., Y.L., Y.S.R., L.P., B.S.H., K.K., S.M.P.-M., S.P., P.A.N., and J.M.C.; experiments in humans, J.E.S. and D.R.R.; patient recruitment and phenotyping, R.E.G., B.S.E., and D.M.M.-M.; statistical analysis, C.L., J.N.S., and J.E.S.; resources, funding, and supervision, S.S.Z. and T.-Y.E.; writing—original draft preparation, T.-Y.E.; writing—review and editing, S.S.Z., T.-Y.E., J.E.S., J.M.C., Y.L., and C.M.D. All authors have read and agreed to the published version of the manuscript.

## Competing interests

The authors declare no competing interests.

## Additional information

[1]Department of Developmental Neurobiology, St. Jude Children's Research Hospital, Memphis, TN 38105, USA. [2]Division of Neuroradiology, Department of Radiology, Hospital of the University of Pennsylvania, Philadelphia, PA 19104, USA. [3]Brain Behavior Laboratory, Neurodevelopment and Psychosis Section, Department of Psychiatry, University of Pennsylvania, Philadelphia, PA 19104, USA. [4]Center for In Vivo Imaging and Therapeutics, St. Jude Children's Research Hospital, Memphis, TN 38105, USA. [5]Max Planck Florida Institute for Neuroscience, Jupiter, FL 33458, USA. [6]Center for Bioimage Informatics, St. Jude Children's Research Hospital, Memphis, TN 38105, USA. [7]Center for Advanced Genome Engineering, St. Jude Children's Research Hospital, Memphis, TN 38105, USA. [8]Department of Medical and Molecular Genetics, Indiana University School of Medicine, Indianapolis, IN 46202, USA. [9]Department of Cell & Molecular Biology, St. Jude Children's Research Hospital, Memphis, TN 38105, USA. [10]Department of Pediatrics, Perelman School of Medicine, University of Pennsylvania, Philadelphia, PA 19104, USA. [11]Division of Human Genetics, Children's Hospital of Philadelphia, Philadelphia, PA 19104, USA. [12]Department of Molecular Medicine, Division of Human Biology and Medical Genetics, Sapienza University, Rome 00185, Italy. [13]Department of Biostatistics, St. Jude Children's Research Hospital, Memphis, TN 38105, USA. [14]Department of Physiology and Biophysics, University of Colorado Anschutz School of Medicine, Aurora, CO 80045, USA. ✉e-mail: stanislav.zakharenko@stjude.org

