## [Transparent Peer Review file · Nature Communications]

Tbx1 haploinsufficiency causes local skull deformity, paraflocculus and flocculus dysplasia, and motor-learning deficit in 22q11.2 deletion syndrome

Corresponding Author: Dr Stanislav Zakharenko

Version 0:

Reviewer comments:

Reviewer #1

(Remarks to the Author)

This paper presents a series of well documented assays that suggest very localized changes in posterior skull volume due to heterozygous deletion of murine orthologues of the genes heterozygously deleted in 22q11.2 Deletion Syndrome in humans. The morphometric analysis using MR imaging in mouse and complimentary MR imaging in humans provides more definitive information of a highly penetrant phenotype of fairly high magnitude in mouse; however, the parallel phenotype in 22q11.2 Deleted humans is less clear. The subsequent data provides evidence that is in some cases consistent with this phenotype modestly disrupting some aspects of cerebellar function, but not cytology. While these data are internally consistent, they are difficult to integrate into a coherent account that supports many of the conclusions reached. Fundamentally, the morphological disruption of the skull surrounding the paraflocculus and flocculus may or may not be related to any of the behavioral or physiological changes described. Additional caution in the presentation and interpretation of much of the data is necessary prior to publication. Finally, the lack of successful rescue by re-expression of Tbx1 does not diminish the apparent influence of Tbx1 on the skull phenotype; however, it does demand far greater caution in drawing any conclusions than what is implied in the Abstract: "This PF/F dysplasia was accompanied by impaired synaptic plasticity and motor learning and caused by haploinsufficiency of the Tbx1 gene encoding a T-box factor." It is simply impossible to get to that conclusion based upon the data presented in this manuscript. There is no doubt that Tbx1 contributes to the skeletal phenotype. Nevertheless, the relationship of this phenotype to cerebellar neuronal and circuit changes detected by the assays described here is tenuous and should be explained far more cautiously before these results are published.

I have the following specific comments on the manuscript:

1. The mouse and human data very different in Figure 1. Please be clearer about this divergence in the Results. The divergence, and its likely meaning, is not commented on accurately until the Discussion.
2. There is a need for more clarity regarding Tbx1 data using nested deletions. The Df(16)2/+ genotype results in phenotypic change that may or may not be statistically different from that for Tbx1 heterozygous deletion at all. The singular mean values amplify what is potentially not actually a statistical difference. Thus, the authors should do statistical tests on phenotypic change between different genotypes: is Tbx1 similar or different statistically from IgDel, DF16 etc. where larger or smaller diffs are seen.
3. It would be useful to know whether all of the genotypes studied were on the same background, and how the mutant alleles were transmitted. The magnitude of the change and variation for 2 of the genotypes: LgDel+ and Df(16)1/+ suggests that this would survive "background" or parent of origin effects. The additional differences might not if the WT and mutants were not rigorously controlled for these variables.
4. The statement "Together, these results suggest that Tbx1 haploinsufficiency is the major driver of the PF/F dysplasia in 22q11DS mouse models" may be an over-interpretation based upon the data as presented. Once some of the analytic issues described above have been addressed, it may be necessary to adjust this statement to better reflect the data.
5. In Figure 3 B and C: it's difficult to see the data points. Filling in the circles, or presenting the data histograms might help

the reader to see more clearly that the genotypes do not differ in their baseline VOR.

6. The VOR data is not well explained, and once again is over-interpreted. There is a very selective genotype-dependent change in the VOR gain that is task specific, and quantitatively divergent in Df(16)1/+ mice versus Tbx1+/- mice. Nevertheless, the authors state at the end of their presentation of the data that “These results suggest that Tbx1 haploinsufficiency disrupts the VOR adaptation. (sic)” This is really not the case. Df(16)1/+ disrupts a specific aspect of VOR adaptation that reflects task demands, and Tbx1 modulates that aspect of the VOR much less substantially. This is not at all what the authors claim.

7. The LTD results, while suggestive of shared, but potentially not identical, disruption in Df(16)1/+vs. Tbx1+/- are not necessarily related to the specific VOR behavioral change reported, except in a very general way. The slice recording indicates that there may be a broad range of circuit or cell autonomous molecular disruptions that are independent of the physical constraints on the paraflocculus/flocculus imposed by the primary skull phenotype—a point that is reinforced by data reported in Figure 5 and the subsequent section of the Results. Thus, these changes versus VOR behavioral changes that could originate in multiple cerebellar regions and circuits, are not formally shown to be related to the primary Df (16)1/+ or Tbx1 changes.

8. The cellular assessment of cytological disruption of development is of somewhat low resolution, and thus should be explained and interpreted cautiously. The results show that there is no discernable change in 1 proliferative, 1 mitotic and 1 apoptotic markers in post-natal cerebellar precursors in Df(16)1/+ mice. The single cell transcriptomics of Tbx1+/- cerebellar samples is consistent with this conclusion; however unless similar

9. The subarcuate fossa phenotype is illustrated with an image for Df161/+, but not Tbx1. It would be helpful to have parallel Tbx1 images.

10. It is not clear why the semicircular canal data is presented. This has been reported before and a causal connection between this change and the cranial skeletal phenotype is not at all certain.

11. The last section on resistance to Tbx1 mediated rescue makes the argument for Tbx1 as the primary driver of many of the phenotypes reported somewhat problematic. Although this may be a limitation of the genotypes or pharmacological agents available for rescue, the formal difficulty introduced into the argument the paper sets forth is substantial. This should be considered carefully against the claims of contiguous gene function for Tbx1 that are made in the paper.

12. The Discussion is far too confident in the assignment of Tbx1 dosage effects on the observed cerebellar structural, cellular and physiological phenotypes. Even the relationship between the apparent skeletal deformity and the behavioral and physiological phenomena reported is not clear. Also, alternative explanations of the data, including disruption of the vestibular inputs to the cerebellum for some of the behavioral changes, should be more thoroughly considered.

13. The use of the word “drastic” which is used throughout the manuscript to describe the magnitude of changes in paraflocculus/flocculus volume doesn’t seem right for these results. It is a substantial reduction in some cases; there is some variation between genotypes, and the mechanistic origins still are uncertain. Perhaps it’s wisest to report the percentage changes in the text and discussion and allow the readers to base their assessment of the phenotypes upon the data.

Reviewer #2

(Remarks to the Author)

Comments:

This paper by Eom et al. reports comprehensive re-examination of brain structures, focusing on specific cerebellar lobules and surrounding bones in 22q11DS subjects as well as a series of rodent models for this microdeletion syndrome. The authors show a previously unrecognized dysplasia of two cerebellar lobules (paraflocculus/flocculus (PF/F)) across the species, the part of the cerebellum responsible for a form of motor learning (vestibulo-ocular reflex (VOR)), and concomitant reduction in size of subarcuate fossa (SF), a part of the petrous temporal bone surrounding the PF/F. Combining an impressive array of brain imaging, genetic, motor learning, cellular-level neuroplasticity assays, and single-nucleus transcriptomics, they conclude that Tbx1 gene deficiency (one of the genes included in the 22q11.2 chromosome region) is primarily responsible for SF deformity through precocious differentiation of chondrocytes to osteoblasts, which then occludes the cerebellar lobule development, resulting in a specific motor learning deficit.

I have one major comment regarding the interpretation of data. I mostly agree with the authors’ prediction that the anatomical structural deficit (skeletal deformity) could limit cerebellar development and later functionality, which is a very novel and provocative idea. However, an alternative explanation is that the neuroplasticity deficit (impaired LTD at the parallel fiber-Purkinje cell synapses) could still be primarily due to aberrant wiring of cerebellar circuitry in 22q11DS models. Although the authors show that Tbx1 expression is restricted specially (rhombic lip) and developmentally (E10) among the cerebellar neuron types and therefore the observed functional deficits (impaired LTD, motor learning) may not be caused by a dysfunction intrinsic to neurons, there is a possibility that cerebellar circuit functions are regulated through afferents incoming from outside the cerebellum, irrespective of maladaptive mechanisms caused by the smaller SF bone. For instance, as shown in Fig.4, the climbing fibers originating from inferior olive in brain stem appears to fail in generating LTD in 22q11DS models, and this could be due to some deficit at the climbing fiber-Purkinje cell synapses possibly caused by Tbx1 deficiency early on during the circuit developmental stage. To argue against this possibility, the authors could check

potential cellular and subcellular problems (e.g., synaptic density at the climbing fiber-Purkinje cell synapses is normal?; # of climbing fiber is normal?; can the authors show that the functional connectivity at these synapses are intact?), and also developmental expression of Tbx1 in the surrounding cerebellar nuclei.

At present, there appears to be a logical leap in the authors' claim that a smaller skeletal structure is causal for neurocircuit functions, unless they show direct evidence that anatomical skeletal size or lobule size correlates with functional performance via functional imaging studies. That said, the current title sounds like a bit of overstatement; they can claim that "Local cerebellar dysplasia and are attributed to Tbx1 deficiency" at best.

Minor suggestions:

- page 3 line 63, Parkinson -> Parkinson's
- In introduction, it may be helpful to clarify that cerebellar skeletal deformity was overlooked or not reported in previous studies on MRI in 22q11DS and the rodent models (Refs. 28-31, 32-33).
- At the beginning of Results, it is helpful to state the rationale of why the cerebellum was chosen as experimental material.
- In Figure 1E, clarify the age used for the study.
- Page 8, line 183, state why mice older than P160 were chosen for analysis.
- On page 9, explanation of experimental paradigm (lines 204-205, 206-207) would better fit in previous section on page 8.
- Page 15, lines 353-354, as mentioned above, it remains unclear whether "the effect of Tbx1 haploinsufficiency on the PF/F phenotype is non-cell-autonomous" or not, unless additional experimental evidence support the authors' claim, so better rephrase for now.
- In the same vein, on page 18, lines 425-427, "a common Tbx1-dependent mechanism" may be an over-interpretation, because the structural deficiency (smaller bone and lobules) and possible addition mechanisms intrinsic to neurons/neural circuit may be distinct, while both are still downstream of Tbx1 deficiency.

Reviewer #3

(Remarks to the Author)

In this paper, Eom et al. address the evidence of structural changes observed in mouse models of 22q11.2 deletion, linked to schizophrenia in humans. They identified a region within the microdeletion, containing the Tbx1 gene, apparently responsible for the reduced volume of the flocculus/paraflocculus (PF/F) region of the cerebellum, the aberrant development of the bone formation surrounding this region, and the defects in vestibulo-ocular adaptations. These findings are supported by several experiments, with appropriate controls and large cohorts. Only the rescue experiments are less convincing. However, I would recommend it for publication with revisions. Specific points below.

One strength of this paper is the genetic approach, with several mouse lines generated to narrow down the Tbx1 gene as crucial for the brain/bone dysplasia, as well as the snRNA-Seq to identify which cell population was most affected by Tbx1 haploinsufficiency. The data suggests that chondrocytes prematurely differentiate into osteoblast as a result of reduced Tbx1 expression, while the cellular composition of the cerebellum is not perturbed and none of the cell types in the cerebellum seem affected. The premature ossification of the SF seems to be the likely explanation for reduced cerebellar volume. This is a plausible explanation, however, there is another possibility that was not explored and should at least be mentioned in the discussion: the climbing fibers originate in the inferior olive and this region, which provides essential input to Purkinje cells and LTD-dependent VOR, was not explored for transcriptional alterations. Furthermore, climbing fibers are in higher number during development and are later reduced to one fiber per Purkinje cell (<https://www.nature.com/articles/ncomms3732>). Is this process unaffected by Tbx1 deficiency? The authors did not address this possibility even though they found deficits in cerebellar LTD, and should acknowledge this caveat adding this evidence. This is important because while bone structure can mechanically affect PF/F volume, it is unclear what disrupts LTD and cerebellar learning underlying oculo-motor adaptations, given that the cell composition, transcriptional profile and migration seem normal in the cerebellum. Additionally, malformation of the semicircular canals in the inner ear could contribute to impaired motor learning by conveying aberrant information to the vestibular pathways and the vestibulo-cerebellum. Although there is a section of the discussion about the inner ear (lines 493-499), this possible link, which would fit with the data and would strengthen the interpretation, was not explained.

Another weakness of the paper is the functional relevance. Oculo-motor deficits are not the most salient feature of individuals with schizophrenia, but the socio-cognitive symptoms are. It would have been nice to see if Tbx1 haploinsufficiency recapitulates also the socio-cognitive phenotype of 22q11.2. This addition would have given the paper more impact, but I think the paper deserves to be published even if the functional focus was solely the oculo-motor adaptation. But claims should be a bit reduced.

In Fig 1 the authors specify that they used both manual and machine-learning-based measurement of regional volumes in MRI data. However, in fig 2 where the author compares the volumes in 16 different mutant lines this is not specified, and also it's not even clear if the volume was measured based on MRI data (the result section doesn't say what method was used and there are no representative MRI images in the fig). The figure title is "Unbiased screen for the gene(s) responsible for paraflocculus/flocculus dysplasia in 22q11DS mice" so it would suggest that the measuring was done through machine learning but it's not explicitly stated. Please specify.

The failure to mimic the phenotype by conditionally silencing Tbx1 levels and to rescue it by conditionally overexpressing it during given developmental stages in my opinion does not invalidate the rest of the results. It is notoriously tricky to pinpoint the exact moment in development when genes are crucial for differentiation and the exact amount of expression needed for

correct gene function. I appreciate the efforts and the transparency of the author in acknowledging this limitation of their study. However, I was wondering if gene silencing/overexpression could be done in cell cultures using progenitors at various stages to narrow down the time point and refine the strategy to be used in vivo. If so, this should be covered in the discussion. The section of the discussion explaining this negative result (lines 517-520) could be improved.

In lines 336-337 the authors write: "The thorough cell-type mapping in the PF/F not only validated the well-established 337 architecture but also underscored the high quality of cellular capture in our data set". While the first expression is perfectly fine, the second expression should be avoided as it may be perceived as "self-praise" while the results section should be very objective.

In lines 336-337 the authors write: "The thorough cell-type mapping in the PF/F not only validated the well-established 337 architecture but also underscored the high quality of cellular capture in our data set". While the first expression is perfectly fine, the second expression should be avoided as it may be perceived as "self-praise" while the results section should be very objective.

Two statements in lines 377 and 378 regarding E7.5-E15.5 being the developmental period of highest Tbx1 expression and E11.5-E18.5 being the period of SCs and SF formation are lacking citations.

Some typos: line 473: "cerebellums" (plural is cerebella); line 189 "increased instead of decreased" (increased instead of decreasing).

Version 1:

Reviewer comments:

Reviewer #1

(Remarks to the Author)

The authors have undertaken a conscientious and thorough review.

Some of the issues with the data in the paper remain: small effect sizes, variability and a sense of over-interpretation of some of the data. The revisions at least provide some clarity for the reader that the overstatements are opinions and speculations of the authors, and not necessarily directly supported by the data provided. This is particularly true for the interpretation of the anomalous VOR gain result, whose functional significance remains a bit obscure, and the statement that Tbx1 is a "major determinant affecting the development of the PF/F-SF-SC complex". The data makes the case that the cranial changes are likely dependent on Tbx1, but the cerebellar neural circuit changes are not well-explained by Tbx1 deletion or function. This could be clarified further at the outset of the discussion.

There are two specific issues in the Discussion that require clarification.

1. The paragraph beginning at line 399 and ending at line 404 is superfluous. It is true that the additional genes listed, although they may influence other aspects of eye movement control, do not seem to affect the VOR in the way that Tbx1 does. That observation, however, does not in anyway reinforce the argument that Tbx1 is a driver of the PF/F dysplasia, with the implication that this is a driver of key aspects of eye movement control. This paragraph should be deleted.
2. The phrase at lines 413-414 that comments on Tbx1 influence on adult hippocampal cell proliferation, differentiation should be eliminated. The reference given is to a study in which Tbx1 was anomalously over-expressed in neural stem cells and thus the physiological relevance of this study is uncertain at best, and at worst misleading. The data in Eom et al support Tbx1 influence on cranial morphogenesis with a modest secondary effect on cerebellar morphology and an unresolved variable impact on some aspects of function. The implication that Tbx1 could influence neural progenitors, proliferation and differentiation promises more than the data can deliver.

Reviewer #2

(Remarks to the Author)

The manuscript by Eom and colleagues has substantially improved after revision, particularly with additional data showing no apparent deficit in synaptic connectivity at Cf-PC and Pf-PC junctions in Df(16)1/+ or Tbx1(+/-) mice. Although this still does not completely rule out the possibility that additional cerebellar circuits or other mechanisms intrinsic to neuronal functions may be involved in the impaired synaptic plasticity and motor learning in these mice, the authors could claim that the observed phenotypes are unlikely to be caused by dysfunction of the major synaptic inputs in cerebellar circuitry. Additionally, given no significant expression of Tbx1 gene in neuronal domains around the cerebellar region during development and adulthood, it may be reasonable to focus on alternative mechanisms in cell types in which this gene is highly expressed. I suggest the authors to include this negative finding (i.e., no obvious neuronal connectivity deficit detected in 22q11DS model and Tbx1/+ mice) in Abstract, such that "However, no overt deficits in neural cell composition, neurogenesis, or synaptic connectivity were identified in the dysplastic PF/F".

Another minor point:

The authors mentioned that they used both males and females for analysis in Methods section, but there is no mention of how many (or percentage) of them were actually used in each experiment, and if sex-specific effects were observed. I recommend clarifying this by including exact numbers of mice used in figure legend, and by reporting sex effect if relevant or where appropriate.

Reviewer #3

(Remarks to the Author)

The revised version of the paper by Eom and colleagues shows improvement compared to the previous version. Indeed, the former version was experimentally thorough, with immense effort put by the authors to narrow down the gene responsible for the skeletal phenotype, to perform measurements to quantify skeletal and brain dimensions, and to assess cytological, transcriptional, synaptic, and functional features of mice carrying these mutation, and even found similarities in mice and humans with deletions affecting similar DNA regions. Methods used are high standard and rigorous. The authors brought together diverse techniques and expertise, which is challenging, creating a nice multi-disciplinary work. However some of the findings were a bit disconnected. All reviewers pointed out a weak link between the skeletal phenotype and the cerebellar LTD defects found in mice with a Tbx1 heterozygous mutation. These two observations appeared to be almost independent, and the defect in cerebellar LTD resulting in mild impairment of VOR adaptation, also seemed weakly linked to Tbx1 deficiency, but was nonetheless observed in these mutants and is an important functional finding. The mechanistic link between these observations and the Tbx1 would have been a plus, but could be pursued as a future direction. The paper brings already an original contribution, which is that Tbx1 is responsible for the reduced volume of the flocculus/paraflocculus (PF/F) region of the cerebellum and the aberrant development of the skull surrounding this region. In this regard, they provide solid evidence to link gene function with phenotype, which is an important contribution. The cerebellar LTD impairment that the authors found could be due to the disruption of some developmental process, but this remains unclear. The important point is that the authors clarified this limitation and carefully interpret this result as unlinked from the skeletal phenotype, and in my opinion they did that by editing the discussion and other parts of the text. They also added more data to address concerns about climbing fiber to Purkinje cells inputs integrity, and more analysis to quantify markers of proliferation, mitosis and apoptosis, confirming no differences between Tbx1 +/- and WT. They clarified quantification methods, provided more details about the animals used and age groups, and now updated the methods properly for the work to be reproduced.

I only have one really minor suggestion for the authors. In the newly added section:

“In contrast, cerebellar (PF/F) cell populations remained stable; no significant shifts (Fig 7h-j, Supplementary Fig. 7h, j) and no change in cell proliferation, cell cycle, or apoptosis was observed between WT and Tbx1 +/- mice (data not shown)”.

Is the “data not shown” available in a public repository? If so, this could be specified. Or, if referring to the data provided in the rebuttal letter, why not include it in the supplements?

Overall, I am satisfied with the revised manuscript and support this work for publication without further revision.

Reviewer #4

(Remarks to the Author)

* I co-reviewed this manuscript with one of the reviewers who provided the listed reports

Version 2:

Reviewer comments:

Reviewer #1

(Remarks to the Author)

I thank the authors for their conscientious and complete effort to modify the manuscript based upon my concerns as well as those of the other reviewers.

Reviewer #2

(Remarks to the Author)

In this revised manuscript, the authors adequately addressed all the concerns I had with the previous version. I am in agreement with the authors' view that potential deficits in neural connectivity around Purkinje cells cannot be ruled out to explain the impaired LTD observed in Tbx1 +/- or Df(16)1/+ mice. I believe this work is now deemed satisfactory to be accepted for publication.

Reviewer #3

(Remarks to the Author)

authors addressed my previous suggestions improving the manuscript.

Reviewer #4

(Remarks to the Author)

REVIEWER COMMENTS

Reviewer #1

This paper presents a series of well documented assays that suggest very localized changes in posterior skull volume due to heterozygous deletion of murine orthologues of the genes heterozygously deleted in 22q11.2 Deletion Syndrome in humans. The morphometric analysis using MR imaging in mouse and complimentary MR imaging in humans provides more definitive information of a highly penetrant phenotype of fairly high magnitude in mouse; however, the parallel phenotype in 22q11.2 Deleted humans is less clear. The subsequent data provides evidence that is in some cases consistent with this phenotype modestly disrupting some aspects of cerebellar function, but not cytology. While these data are internally consistent, they are difficult to integrate into a coherent account that supports many of the conclusions reached. Fundamentally, the morphological disruption of the skull surrounding the paraflocculus and flocculus may or may not be related to any of the behavioral or physiological changes described. Additional caution in the presentation and interpretation of much of the data is necessary prior to publication. Finally, the lack of successful rescue by re-expression of *Tbx1* does not diminish the apparent influence of *Tbx1* on the skull phenotype; however, it does demand far greater caution in drawing any conclusions than what is implied in the Abstract: “This PF/F dysplasia was accompanied by impaired synaptic plasticity and motor learning and caused by haploinsufficiency of the *Tbx1* gene encoding a T-box factor.” It is simply impossible to get to that conclusion based upon the data presented in this manuscript. There is no doubt that *Tbx1* contributes to the skeletal phenotype. Nevertheless, the relationship of this phenotype to cerebellar neuronal and circuit changes detected by the assays described here is tenuous and should be explained far more cautiously before these results are published.

We thank Reviewer 1 for this comment and for agreeing that *Tbx1* haploinsufficiency contributes to the skeletal phenotype. We have softened the interpretations of our results by modifying the Title, Abstract, and other parts of the manuscript.

I have the following specific comments on the manuscript:

1. The mouse and human data very different in Figure 1. Please be clearer about this divergence in the Results. The divergence, and its likely meaning, is not commented on accurately until the Discussion.

The paraflocculus (PF) and flocculus (F) are smaller in humans with 22q11DS and in 22q11DS mice. The difference is only in the degree of this phenotype. We have added a brief explanation of this difference in the Results (lines 116-118) as suggested. However, we felt that a more comprehensive explanation of this difference should be presented in the Discussion.

2. There is a need for more clarity regarding *Tbx1* data using nested deletions. The *Df(16)2/+* genotype results in phenotypic change that may or may not be statistically different from that for *Tbx1* heterozygous deletion at all. The singular mean values amplify what is potentially not actually a statistical difference. Thus, the authors should do statistical tests on phenotypic change between different genotypes: is *Tbx1* similar or different statistically from *IgDel*, *DF16* etc. where larger or smaller diffs are seen.

We thank the Reviewer for bringing up this point. We have now included statistical analyses of the nested deletions. We performed a *t*-test considering the mean volume of each nested deletion (denoted V_{Mut}) and its matched wild-type volume [denoted $V_{WT}(Mut)$] in the Results (lines 150-161). We added the following text: To further confirm that *Tbx1* haploinsufficiency is a major contributor to the PF/F dysplasia in 22q11DS mice, we tested if the PF/F reduction in *Tbx1*^{+/-} mice differs from that in mutants with

multigene deletions containing the *Tbx1* gene by using two-sample *t*-test. The reduction of the PF/F volume in *Tbx1*^{+/-} mice was in line with differences in *Del(3.0 Mb)/+* ($t_{48.59}=-0.13$, $p=0.9$), *Df(16)3/+* ($t_{6.35}=-0.08$, $p=0.94$), and *Df(16)4/+* ($t_{11.12}=0.99$, $p=0.34$) mice, though we detected a significant difference from that in *LgDel/+* ($t_{25.98}=2.34$, $p=0.03$) and *Df(16)1/+* mice ($t_{48.62}=3.15$, $p=0.003$). The differences observed in *LgDel/+* mice and *Df(16)1/+* mice was due to variability in WT littermate controls, as the mean PF/F volumes in these mutant samples were not significantly smaller than that in *Tbx1*^{+/-} mice [two-sample *t*-test; *LgDel/+* ($t_{17.43}=0.31$, $p=0.76$) and *Df(16)1/+* ($t_{21.37}=1.21$, $p=0.24$)]. The PF/F dysplasia in *Tbx1*^{+/-} mice was more severe than in mice with the *Df(16)2/+* deletion, which does not contain the *Tbx1* gene ($t_{12.99}=2.79$, $p=0.02$). The mean PF/F volumes in *Tbx1*^{+/-} mice were significantly smaller than that in *Df(16)2/+* mice ($t_{11.07}=4.07$, $p=0.002$). Together, these data suggest that *Tbx1* haploinsufficiency is a major contributor to the PF/F reduction in 22q11DS mouse models.

3. It would be useful to know whether all of the genotypes studied were on the same background, and how the mutant alleles were transmitted. The magnitude of the change and variation for 2 of the genotypes: *LgDel*⁺ and *Df(16)1/+* suggests that this would survive “background” or parent of origin effects. The additional differences might not if the WT and mutants were not rigorously controlled for these variables.

The mouse strains used in this study were backcrossed onto the C57BL/6J background for several generations, except *LgDel*⁺ mice, which were maintained on the C57BL/6N background. The mutated alleles were transmitted paternally or maternally, and no difference in the PF/F volume was found in either transmission. The *LgDel* mutation was transmitted paternally. We have added this information to the Methods (Animals).

4. The statement “Together, these results suggest that *Tbx1* haploinsufficiency is the major driver of the PF/F dysplasia in 22q11DS mouse models” may be an over-interpretation based upon the data as presented. Once some of the analytic issues described above have been addressed, it may be necessary to adjust this statement to better reflect the data.

See response to comment #2.

5. In Figure 3 B and C: it’s difficult to see the data points. Filling in the circles, or presenting the data histograms might help the reader to see more clearly that the genotypes do not differ in their baseline VOR.

We have modified Fig. 3b and 3c accordingly.

6. The VOR data is not well explained, and once again is over-interpreted. There is a very selective genotype-dependent change in the VOR gain that is task specific, and quantitatively divergent in *Df(16)1/+* mice versus *Tbx1*^{+/-} mice. Nevertheless, the authors state at the end of their presentation of the data that “These results suggest that *Tbx1* haploinsufficiency disrupts the VOR adaptation. (sic)” This is really not the case. *Df(16)1/+* disrupts a specific aspect of VOR adaptation that reflects task demands, and *Tbx1* modulates that aspect of the VOR much less substantially. This is not at all what the authors claim.

In our original submission, we indicated that “We also tested VOR after opposite- and same-direction mismatch training in *Tbx1*^{+/-} mice and found a deficit of VOR gain for the opposite-direction training

context, though to a lesser extent than that in *Df(16)1/+* mice (Fig. 3e).” We have now modified the end of that data presentation sentence (lines 195-196).

7. The LTD results, while suggestive of shared, but potentially not identical, disruption in *Df(16)1/+*-vs. *Tbx1^{+/-}* are not necessarily related to the specific VOR behavioral change reported, except in a very general way. The slice recording indicates that there may be a broad range of circuit or cell autonomous molecular disruptions that are independent of the physical constraints on the paraflocculus/flocculus imposed by the primary skull phenotype—a point that is reinforced by data reported in Figure 5 and the subsequent section of the Results. Thus, these changes versus VOR behavioral changes that could originate in multiple cerebellar regions and circuits, are not formally shown to be related to the primary *Df(16)1/+* or *Tbx1* changes.

Based on literature, we posited that gain-increase VOR learning depends on climbing fiber (Cf)-induced LTD and gain-decrease learning relies on LTP at parallel fiber (Pf)–Purkinje cell (PC) synapses (lines 199-204). Therefore, the fact that *Tbx1* haploinsufficiency selectively affects VOR gain increase and Pf-PC LTD but no gain decrease and Pf-PC LTP is interesting, and we wanted to emphasize it. However, we agree with the Reviewer that the gain-increase VOR may depend on multiple circuits and have, therefore, adjusted the Discussion accordingly (lines 392-398).

8. The cellular assessment of cytological disruption of development is of somewhat low resolution, and thus should be explained and interpreted cautiously. The results show that there is no discernable change in 1 proliferative, 1 mitotic and 1 apoptotic markers in post-natal cerebellar precursors in *Df(16)1/+* mice. The single cell transcriptomics of *Tbx1^{+/-}* cerebellar samples is consistent with this conclusion; however unless similar

Initially, we displayed the immunostaining data (Ki67, PH3, c-casp3) over a large area that included parts of the cerebellum with the PF/F. We also converted PH3 and c-casp3 images to black and white format for a better view in the print version. We have now magnified these figures, with the focus on the PF/F (**Fig. 5a, d, g**) for better display of these data.

We assume that Reviewer 1 meant to comment on congruency between immunohistochemistry and snRNA-seq data. Our immunohistochemistry data showed no difference in the expression of cell proliferation, mitotic, or apoptotic markers in the PF/F of WT mice and *Tbx1^{+/-}* mice. We have now performed differential expression analysis of *Mki67*, *Phc3*, and *Casp3* markers in the PF/F and found no difference between WT and *Tbx1^{+/-}* mice. We also tested the enrichment of the cell cycling gene set (94 cycling-related genes in the Seurat package) and the apoptosis gene set (85 apoptosis-related genes in KEGG) and the result showed no differences between the WT and *Tbx1^{+/-}* mice (see below). Thus, snRNA-seq data are consistent with the immunohistochemistry data. We have added a short description of these data (lines 331-333). We thank Reviewer 1 for suggesting these analyses.

Fig. The expression of cell proliferation, mitotic, and apoptotic markers in the PF/F does not differ between WT mice and *Tbx1*^{+/-} mice. **a** $p=0.39$, adjusted p -value=1, $\log_2FC=-0.05$, 41.9% of cells in *Tbx1*^{+/-} mice show this gene expression >0, 41.8% of cells in WT mice show this gene expression >0. **b** $p=0.11$ adjusted p -value=1, $\log_2FC=-0.03$, 17.7% of cells in *Tbx1*^{+/-} mice show this gene expression >0, 19.2% of cells in WT mice show this gene expression >0. **c** $p=0.003$, adjusted p -value=1, $\log_2FC=-0.02$, 11.3% of cells in *Tbx1*^{+/-} mice show this gene expression >0, 12.2% of cells in WT mice show this gene expression >0. **d** UMAP of PF/F of WT and *Tbx1*^{+/-} mice colored by predicted cell cycle phase from Seurat. The proportion of cycling cells showed no significant difference between the two groups. **e** Quantification of the UMAP in **d**, $p=0.01$, $\log_2FC=0.08$, 75.4% of cells in *Tbx1*^{+/-} mice show this gene expression >0, 78.9% of cells in WT mice show this expression >0. **f** $p=0.05$, $\log_2FC=-0.05$, 2.53% of cells in *Tbx1*^{+/-} mice show this gene expression >0, 2.68% of cells in WT mice show this expression >0. Significance cutoff: adjusted p -value <0.05, FC>1.5, $\log_2FC >0.58$, proportion of positive cells >10%. n.s., not significant.

9. The subarcuate fossa phenotype is illustrated with an image for *Df161*+, but not *Tbx1*. It would be helpful to have parallel *Tbx1* images.

We have added subarcuate fossa images of *Tbx1* mutants and WT control mice in **Fig. 6c-c'**, **d-d'**.

10. It is not clear why the semicircular canal data is presented. This has been reported before and a causal connection between this change and the cranial skeletal phenotype is not at all certain.

Previous studies on semicircular canals (SCs) were based on homozygous, conditional *Tbx1* deletion (*Cre;Tbx1^{flox/flox}*) or conditional deletion of *Cre* with *Tbx1^{flox/-}*. We now show that the SC phenotype is present in *Tbx1^{+/-}* mice. Furthermore, the SCs and subarcuate fossa (SF) are spatially interconnected: the SF forms through the arc of the anterior semicircular canal (ASC), and the sizes of these two structures are correlated during development (PMID: 16950498). In addition, the SF cavity is structurally surrounded by the lateral and posterior SCs. Therefore, the SCs and SF are spatially interlinked, and we felt compelled to describe SCs in our work.

11. The last section on resistance to *Tbx1* mediated rescue makes the argument for *Tbx1* as the primary driver of many of the phenotypes reported somewhat problematic. Although this may be a limitation of the genotypes or pharmacological agents available for rescue, the formal difficulty introduced into the argument the paper sets forth is substantial. This should be considered carefully against the claims of contiguous gene function for *Tbx1* that are made in the paper.

We have modified our description of the limitations of our genetic or pharmacologic approaches in the Discussion as follows: “The genetic approach to identifying a cell-specific role of *Tbx1* and to mimic and rescue the phenotypes by using the Cre-lox system *in vivo* has failed, which could be due to several limitations that hinder the interpretation of those results. Manipulation of *Tbx1* expression in certain Cre lines (*Mesp1^{Cre}*, *Twist2^{Cre}*, and *Foxg1^{Cre}*) leads to lethality. Furthermore, the negative results may be caused by incomplete gene excision of the floxed loci or inappropriate timing of recombination. *Tbx1* manipulation in these lineages also may not specifically affect chondrocyte-to-osteoblast differentiation. Thus, more refined tools or approaches to control *Tbx1* dosage in a spatiotemporal manner *in vivo* will be needed to help understand the roles of *Tbx1* during cerebellar development.” (lines 488-496).

12. The Discussion is far too confident in the assignment of *Tbx1* dosage effects on the observed cerebellar structural, cellular and physiological phenotypes. Even the relationship between the apparent skeletal deformity and the behavioral and physiological phenomena reported is not clear. Also, alternative explanations of the data, including disruption of the vestibular inputs to the cerebellum for some of the behavioral changes, should be more thoroughly considered.

We now discuss our results more cautiously. With respect to the vestibular inputs, our VOR data argues against the disruption of vestibular inputs to the cerebellum in 22q11DS mice. **Fig. 3b, c** shows normal baseline VOR gain in *Df(16)1/+* models of 22q11DS and *Tbx1^{+/-}* mice. We describe “...a similar baseline VOR performance (1 Hz; 0.69 ± 0.06 and 0.72 ± 0.12 for WT and *Df(16)1/+* mice, respectively, and 0.68 ± 0.1 and 0.66 ± 0.08 for WT and *Tbx1^{+/-}* mice, respectively).” Thus, the VOR adaptation deficit we describe in these mutants (**Fig. 3d, e**) is unlikely caused by the aberrant vestibular inputs to the cerebellum but rather by abnormal plasticity mechanisms. A possible candidate for these mechanisms is abnormal Pf-PC LTD described in **Fig. 4a-d**.

13. The use of the word “drastic” which is used throughout the manuscript to describe the magnitude of changes in paraflocculus/flocculus volume doesn’t seem right for these results. It is a substantial reduction in some cases; there is some variation between genotypes, and the mechanistic origins still are uncertain. Perhaps it’s wisest to report the percentage changes in the text and discussion and allow the readers to base their assessment of the phenotypes upon the data.

We appreciate this suggestion and have now removed the word “drastic” throughout the manuscript and described the phenotype in terms of percentage changes.

Reviewer #2

1. This paper by Eom et al. reports comprehensive re-examination of brain structures, focusing on specific cerebellar lobules and surrounding bones in 22q11DS subjects as well as a series of rodent models for this microdeletion syndrome. The authors show a previously unrecognized dysplasia of two cerebellar lobules (paraflocculus/flocculus (PF/F)) across the species, the part of the cerebellum responsible for a form of motor learning (vestibulo-ocular reflex (VOR)), and concomitant reduction in size of subarcuate fossa (SF), a part of the petrous temporal bone surrounding the PF/F. Combining an impressive array of brain imaging, genetic, motor learning, cellular-level neuroplasticity assays, and single-nucleus transcriptomics, they conclude that *Tbx1* gene deficiency (one of the genes included in the 22q11.2 chromosome region) is primarily responsible for SF deformity through precocious differentiation of chondrocytes to osteoblasts, which then occludes the cerebellar lobule development, resulting in a specific motor learning deficit.

I have one major comment regarding the interpretation of data. I mostly agree with the authors' prediction that the anatomical structural deficit (skeletal deformity) could limit cerebellar development and later functionality, which is a very novel and provocative idea. However, an alternative explanation is that the neuroplasticity deficit (impaired LTD at the parallel fiber-Purkinje cell synapses) could still be primarily due to aberrant wiring of cerebellar circuitry in 22q11DS models. Although the authors show that *Tbx1* expression is restricted specially (rhombic lip) and developmentally (E10) among the cerebellar neuron types and therefore the observed functional deficits (impaired LTD, motor learning) may not be caused by a dysfunction intrinsic to neurons, there is a possibility that cerebellar circuit functions are regulated through afferents incoming from outside the cerebellum, irrespective of maladaptive mechanisms caused by the smaller SF bone. For instance, as shown in Fig.4, the climbing fibers originating from inferior olive in brain stem appears to fail in generating LTD in 22q11DS models, and this could be due to some deficit at the climbing fiber-Purkinje cell synapses possibly caused by *Tbx1* deficiency early on during the circuit developmental stage. To argue against this possibility, the authors could check potential cellular and subcellular problems (e.g., synaptic density at the climbing fiber-Purkinje cell synapses is normal?; # of climbing fiber is normal?; can the authors show that the functional connectivity at these synapses are intact?), and also developmental expression of *Tbx1* in the surrounding cerebellar nuclei.

We thank Reviewer 2 for this positive review of our manuscript and for suggesting an alternative hypothesis. We checked the functional connectivity at Cf-PC synapses in *Df(16)1/+* mice and *Tbx1*^{+/-} mice. We found that Cf stimulation evoked complex spikes of similar amplitudes in PCs in mutants and their respective WT controls. These data argue against the notion that Cf inputs are compromised in 22q11DS mice. No difference was also found in functional connectivity at and Pf-PC synapses. We describe these new data in **Supplementary Fig. 4** and in lines 211-214. We also modified the Discussion accordingly (see response 7, Reviewer 1).

There is a general agreement that *Tbx1* is expressed at very low levels in the brain. We emphasized this point (lines 408-413). We also discussed *Tbx1* expression during development in the pharyngeal endoderm, mesodermal core of the pharyngeal arches, head mesenchyme, secondary heart field, sclerotome, and otocyst. Among these regions, head mesenchyme and otocyst are the most adjacent to the cerebellum. In mice, *Tbx1* expression was detected exclusively in the endothelial cells of the brain vasculature, not in other brain cell types (PMID: 16684884). In the developing brain, *Tbx1* was undetectable during early embryonic stages but robustly expressed at postnatal day (P) 25 in the forebrain (PMID: 16684884, PMID: 14614146) and was not found in the developing hindbrain (PMID: 24357327), indicating limited or very low expression of *Tbx1* in the developing brain.

2. At present, there appears to be a logical leap in the authors' claim that a smaller skeletal structure is causal for neurocircuit functions, unless they show direct evidence that anatomical skeletal size or lobule size correlates with functional performance via functional imaging studies. That said, the current title sounds like a bit of overstatement; they can claim that "Local cerebellar dysplasia and are attributed to *Tbx1* deficiency" at best.

We have now changed the title to "*Tbx1* haploinsufficiency causes local skull deformity, paraflocculus and flocculus dysplasia, and motor-learning deficit in 22q11.2 deletion syndrome," which we believe is in keeping with the intent of Reviewer 2.

Minor suggestions:

3) page 3 line 63, Parkinson -> Parkinson's

Corrected.

4) In introduction, it may be helpful to clarify that cerebellar skeletal deformity was overlooked or not reported in previous studies on MRI in 22q11DS and the rodent models (Refs. 28-31, 32-33).

We have now added this information to the Introduction (lines 65-66).

5) At the beginning of Results, it is helpful to state the rationale of why the cerebellum was chosen as experimental material.

We have now clarified this point (lines 83-84). We chose the cerebellum because the cerebellar lobules PF/F were the only brain regions that were substantially reduced in 22q11DS mice.

6) In Figure 1E, clarify the age used for the study.

We have now added the age "2-month-old" in the **Fig. 1e** legend.

7) Page 8, line 183, state why mice older than P160 were chosen for analysis.

We chose mice older than P160 for the VOR test for conceptual and technical reasons. Conceptually, we wanted to avoid developmental stages when the PF/F is not fully established. Technically, we had to transport the mice from St. Jude (Memphis, TN) to Max Planck Florida Institute (Jupiter, FL) for VOR testing. This was associated with a long acclimation period and all other vagaries associated with animal transport between institutions.

8) On page 9, explanation of experimental paradigm (lines 204-205, 206-207) would better fit in previous section on page 8.

We respectfully disagree, as we wrote this text as the rationale for performing LTD/LTP experiments, which we describe immediately thereafter.

9) Page 15, lines 353-354, as mentioned above, it remains unclear whether "the effect of *Tbx1* haploinsufficiency on the PF/F phenotype is non-cell-autonomous" or not, unless additional experimental evidence support the authors' claim, so better rephrase for now.

We have now modified the text as suggested.

10) In the same vein, on page 18, lines 425-427, “a common Tbx1-dependent mechanism” may be an over-interpretation, because the structural deficiency (smaller bone and lobules) and possible addition mechanisms intrinsic to neurons/neural circuit may be distinct, while both are still downstream of Tbx1 deficiency.

We have now modified the text accordingly.

Reviewer #3

1. In this paper, Eom et al. address the evidence of structural changes observed in mouse models of 22q11.2 deletion, linked to schizophrenia in humans. They identified a region within the microdeletion, containing the Tbx1 gene, apparently responsible for the reduced volume of the flocculus/paraflocculus (PF/F) region of the cerebellum, the aberrant development of the bone formation surrounding this region, and the defects in vestibulo-ocular adaptations. These findings are supported by several experiments, with appropriate controls and large cohorts. Only the rescue experiments are less convincing. However, I would recommend it for publication with revisions. Specific points below.

One strength of this paper is the genetic approach, with several mouse lines generated to narrow down the Tbx1 gene as crucial for the brain/bone dysplasia, as well as the snRNA-Seq to identify which cell population was most affected by Tbx1 haploinsufficiency. The data suggests that chondrocytes prematurely differentiate into osteoblast as a result of reduced Tbx1 expression, while the cellular composition of the cerebellum is not perturbed and none of the cell types in the cerebellum seem affected. The premature ossification of the SF seems to be the likely explanation for reduced cerebellar volume. This is a plausible explanation, however, there is another possibility that was not explored and should at least be mentioned in the discussion: the climbing fibers originate in the inferior olive and this region, which provides essential input to Purkinje cells and LTD-dependent VOR, was not explored for transcriptional alterations. Furthermore, climbing fibers are in higher number during development and are later reduced to one fiber per Purkinje cell (<https://www.nature.com/articles/ncomms3732>). Is this process unaffected by Tbx1 deficiency? The authors did not address this possibility even though they found deficits in cerebellar LTD, and should acknowledge this caveat adding this evidence. This is important because while bone structure can mechanically affect PF/F volume, it is unclear what disrupts LTD and cerebellar learning underlying oculo-motor adaptations, given that the cell composition, transcriptional profile and migration seem normal in the cerebellum. Additionally, malformation of the semicircular canals in the inner ear could contribute to impaired motor learning by conveying aberrant information to the vestibular pathways and the vestibulo-cerebellum. Although there is a section of the discussion about the inner ear (lines 493-499), this possible link, which would fit with the data and would strengthen the interpretation, was not explained.

We thank Reviewer 3 for the positive feedback of our manuscript and for proposing alternative mechanisms. Regarding a possible Cf deficiency in 22q11DS mice, please see response 1, Reviewer 2. We have added new data (**Supplementary Fig. 4**) that show that Cf-PC and Pf-PC synaptic responses are not affected in 22q11DS mice or *Tbx1*^{+/-} mice.

Reviewer 3 is correct that malformation of the SCs may contribute to impaired motor learning. However, **Fig. 3b, c** show that the baseline VOR performance is normal in 22q11DS and *Tbx1*^{+/-} mice, suggesting

that the vestibular component of oculomotor control is not impaired in these animals. We argue that VOR adaptation (motor learning) is deficient in these mutants (**Fig. 3d, e**), and this deficiency is not caused by the aberrant vestibular pathway but rather by deficient synaptic plasticity in cerebellar circuits. One possible candidate for this mechanism is deficient Pf-PC LTD, which we describe in **Fig. 4**.

We are grateful to Reviewer 3 for bringing to our attention the paper regarding Cf changes during development. We now cite it in our manuscript.

2. Another weakness of the paper is the functional relevance. Oculo-motor deficits are not the most salient feature of individuals with schizophrenia, but the socio-cognitive symptoms are. It would have been nice to see if *Tbx1* haplo-insufficiency recapitulates also the socio-cognitive phenotype of 22q11.2. This addition would have given the paper more impact, but I think the paper deserves to be published even if the functional focus was solely the oculo-motor adaptation. But claims should be a bit reduced.

Our unbiased MRI of brain volumes identified a specific and substantial reduction in the PF/F volume in 22q11DS mice. Because these cerebellar lobules are responsible for VOR, we chose to focus on this behavior. This is a novel behavioral deficit observed in these mutants. Furthermore, the socio-cognitive deficits in 22q11DS and *Tbx1*^{+/-} mice have been described. For instance, Hiramoto et al., (PMID: 34737458) have shown deficits in cognitive function in *Tbx1*^{+/-} mice, including that in spatial memory acquisition and simple discrimination. It has been also shown that *Tbx1*^{+/-} mice have impaired sensorimotor gating, as measured by prepulse inhibition (PPI) of the startle response (PMID: 16684884). We have now clarified this point in the Discussion (lines 419-423).

3. In Fig 1 the authors specify that they used both manual and machine-learning-based measurement of regional volumes in MRI data. However, in fig 2 where the author compares the volumes in 16 different mutant lines this is not specified, and also it's not even clear if the volume was measured based on MRI data (the result section doesn't say what method was used and there are no representative MRI images in the fig). The figure title is "Unbiased screen for the gene(s) responsible for paraflocculus/flocculus dysplasia in 22q11DS mice" so it would suggest that the measuring was done through machine learning but it's not explicitly stated. Please specify.

We have now clarified the approach used in the Methods (lines 731-733) as follows: "All volumes were measured using manual segmentation of the MRI data, except the data from the machine-learning-based measurement in **Supplementary Fig. 1g, h**." We now refer to data in **Fig. 2** as an unbiased screen because MRI measurements were performed without prior knowledge of the causal gene.

4. The failure to mimic the phenotype by conditionally silencing *Tbx1* levels and to rescue it by conditionally overexpressing it during given developmental stages in my opinion does not invalidate the rest of the results. It is notoriously tricky to pinpoint the exact moment in development when genes are crucial for differentiation and the exact amount of expression needed for correct gene function. I appreciate the efforts and the transparency of the author in acknowledging this limitation of their study. However, I was wondering if gene silencing/overexpression could be done in cell cultures using progenitors at various stages to narrow down the time point and refine the strategy to be used *in vivo*. If so, this should be covered in the discussion. The section of the discussion explaining this negative result (lines 517-520) could be improved.

We appreciate the Reviewer's understanding and acknowledgement of the difficulty of manipulating the *Tbx1* level in a time- and cell type-specific manner using the Cre-lox approach. We also believe that *in vitro* experiments might not fully recapitulate the *in vivo* conditions, especially in such a specialized area

as the PF/F surrounded by complex bone structures, such as the SF and SCs. As suggested, we have clarified the discussion of these data (lines 488-496).

5. In lines 336-337 the authors write: “The thorough cell-type mapping in the PF/F not only validated the well-established architecture but also underscored the high quality of cellular capture in our data set”. While the first expression is perfectly fine, the second expression should be avoided as it may be perceived as “self-praise” while the results section should be very objective.

We apologize for this oversight and have now removed this text.

6. Two statements in lines 377 and 378 regarding E7.5-E15.5 being the developmental period of highest Tbx1 expression and E11.5-E18.5 being the period of SCs and SF formation are lacking citations.

We have now added the appropriate citations.

7. Some typos: line 473: “cerebellums” (plural is cerebella); line 189 “increased instead of decreased” (increased instead of decreasing).

We have now corrected those typos.

Reviewer #1

1. The authors have undertaken a conscientious and thorough review. Some of the issues with the data in the paper remain: small effect sizes, variability and a sense of over-interpretation of some of the data. The revisions at least provide some clarity for the reader that the overstatements are opinions and speculations of the authors, and not necessarily directly supported by the data provided. This is particularly true for the interpretation of the anomalous VOR gain result, whose functional significance remains a bit obscure, and the statement that *Tbx1* is a "major determinant affecting the development of the PF/F–SF–SC complex". The data makes the case that the cranial changes are likely dependent on *Tbx1*, but the cerebellar neural circuit changes are not well-explained by *Tbx1* deletion or function. This could be clarified further at the outset of the discussion.

We thank Reviewer 1 for these constructive comments. We respectfully disagree that our manuscript deals with an issue of small effect sizes. We show that the paraflocculus and flocculus (PF/F) (Fig. 1c-e) and the skeletal subarcuate fossa (SF) (Fig. 6e) are reduced by 50%-70% in mouse models of 22q11 deletion syndrome (22q11DS). This is in stark contrast with the 10%-15% structural changes typically reported in 22q11DS. We also report a substantial functional change in vestibulo-ocular reflex (VOR) gain and PF/F synaptic plasticity in 22q11DS mice. Instead of the increase in VOR gain, 22q11DS mice show a decrease (Fig. 3d). Similarly, instead of long-term depression (LTD), 22q11DS mice show long-term potentiation (LTP) at parallel fiber synapses (Fig. 4b). All these phenotypes are far from subtle.

During the first round of revision, we modified our conclusions in keeping with the data and toned down the interpretation of our results. However, our data clearly show that *Tbx1* haploinsufficiency leads to PF/F dysplasia (Fig. 2b), SF deformity (Fig. 6f), and malformed semicircular canals (SCs) (Fig. 6l-q). Given that PF/F, SF, and SCs are adjacent, interdependent structures and the reduction of PF/F and SF in 22q11DS and *Tbx1*^{+/-} mutants are strongly correlated (Fig. 6j), our conclusion that *Tbx1* is a major determinant affecting the development of the PF/F–SF–SCs complex is supported by our data. We not only acknowledge but also suggest that *Tbx1*-mediated mechanisms affect the PF/F–SF–SCs complex indirectly through the precocious differentiation of chondrocytes to osteoblasts and SF deformity. We also acknowledge that the exact mechanisms by which *Tbx1* affects the individual components of this complex remain to be elucidated in future studies. Nonetheless, we have deleted the above-mentioned phrase in keeping with the reviewer's judgement. We have also further discussed the VOR gain phenotype and cerebellar LTD defect in 22q11DS mutants (lines 388-396).

2. There are two specific issues in the Discussion that require clarification. The paragraph beginning at line 399 and ending at line 404 is superfluous. It is true that the additional genes listed, although they may influence other aspects of eye movement control, do not seem to affect the VOR in the way that *Tbx1* does. That observation, however, does not in anyway reinforce the argument that *Tbx1* is a driver of the PF/F dysplasia, with the implication that this is a driver of key aspects of eye movement control. This paragraph should be deleted.

We have deleted this paragraph as suggested.

3. The phrase at lines 413-414 that comments on *Tbx1* influence on adult hippocampal cell proliferation, differentiation should be eliminated. The reference given is to a study in which *Tbx1* was anomalously over-expressed in neural stem cells and thus the physiological relevance of this study is uncertain at best, and at worst misleading. The data in Eom et al support *Tbx1* influence on cranial morphogenesis with a modest secondary effect on cerebellar morphology and an unresolved variable impact on some aspects of function. The implication that *Tbx1* could influence neural progenitors, proliferation and differentiation promises more than the data can deliver.

We have deleted this phrase as requested.

Reviewer #2

1. The manuscript by Eom and colleagues has substantially improved after revision, particularly with additional data showing no apparent deficit in synaptic connectivity at Cf-PC and Pf-PC junctions in *Df(16)1/+* or *Tbx1(+/-)* mice. Although this still does not completely rule out the possibility that additional cerebellar circuits or other mechanisms intrinsic to neuronal functions may be involved in the impaired synaptic plasticity and motor learning in these mice, the authors could claim that the observed phenotypes are unlikely to be caused by dysfunction of the major synaptic inputs in cerebellar circuitry. Additionally, given no significant expression of *Tbx1* gene in neuronal domains around the cerebellar region during development and adulthood, it may be reasonable to focus on alternative mechanisms in cell types in which this gene is highly expressed. I suggest the authors to include this negative finding (i.e., no obvious neuronal connectivity deficit detected in 22q11DS model and *Tbx1/+* mice) in Abstract, such that "However, no overt deficits in neural cell composition, neurogenesis, or synaptic connectivity were identified in the dysplastic PF/F".

We appreciate Reviewer 2's enthusiasm and suggestion to describe negative findings in the Abstract. Currently, the Abstract includes information about neural cell composition and neurogenesis. Although we did not observe overt deficits in two major synaptic inputs at the Cf-PC and Pf-PC projections in *Df(16)1/+* or *Tbx1^{+/-}* mice, we must exercise caution and acknowledge that we cannot rule out potential deficits in other neuronal cell type connections to Purkinje cells (PCs) (e.g., basket cells, stellate cells) or in synaptic connections of PCs to neurons in the brain stem (e.g., vestibular nuclei) or deep cerebellar nuclei (e.g., dentate nuclei). We briefly described the involvement of multiple plasticity mechanisms in VOR adaptation in the Discussion (lines 391-403). Given that we found impaired LTD synaptic plasticity in the dysplastic PF/F, further investigations of PC function and synaptic connections could be valuable as a future direction. Therefore, while we cautiously specify that we observed no overt deficits in synaptic connectivity at Cf-PC and Pf-PC projections, we prefer to avoid using the broad term "synaptic connectivity" in the Abstract.

2. Another minor point: The authors mentioned that they used both males and females for analysis in Methods section, but there is no mention of how many (or percentage) of them were actually used in each experiment, and if sex-specific effects were observed. I recommend clarifying this by including exact numbers of mice used in figure legend, and by reporting sex effect if relevant or where appropriate.

To test for sex-specific effects, we initially examined sex-dependent differences in the PF/F volumes in the 22q11DS and *Tbx1^{+/-}* mouse models. The data are provided in Supplementary Figs. 1a and 3d, respectively. Because no sex-specific effects were detected, we proceeded with both males and females for further analyses in the rodent study. This clarification is included in the Methods (line 681-682). In the human study, we provide the number of each sex, and the effects of sex on data analyses is included in the Methods (line 741-742, 746-747). We have also provided data showing no sex-specific effects on the PF/F volume in typically developing (TD) human subjects or those with 22q11DS in Supplementary Fig. 2d.

Reviewer #3

The revised version of the paper by Eom and colleagues shows improvement compared to the previous version. Indeed, the former version was experimentally thorough, with immense effort put by the

authors to narrow down the gene responsible for the skeletal phenotype, to perform measurements to quantify skeletal and brain dimensions, and to assess cytological, transcriptional, synaptic, and functional features of mice carrying these mutation, and even found similarities in mice and humans with deletions affecting similar DNA regions. Methods used are high standard and rigorous. The authors brought together diverse techniques and expertise, which is challenging, creating a nice multi-disciplinary work. However some of the findings were a bit disconnected. All reviewers pointed out a weak link between the skeletal phenotype and the cerebellar LTD defects found in mice with a *Tbx1* heterozygous mutation. These two observations appeared to be almost independent, and the defect in cerebellar LTD resulting in mild impairment of VOR adaptation, also seemed weakly linked to *Tbx1* deficiency, but was nonetheless observed in these mutants and is an important functional finding. The mechanistic link between these observations and the *Tbx1* would have been a plus, but could be pursued as a future direction. The paper brings already an original contribution, which is that *Tbx1* is responsible for the reduced volume of the flocculus/paraflocculus (PF/F) region of the cerebellum and the aberrant development of the skull surrounding this region. In this regard, they provide solid evidence to link gene function with phenotype, which is an important contribution. The cerebellar LTD impairment that the authors found could be due to the disruption of some developmental process, but this remains unclear. The important point is that the authors clarified this limitation and carefully interpret this result as unlinked from the skeletal phenotype, and in my opinion they did that by editing the discussion and other parts of the text. They also added more data to address concerns about climbing fiber to Purkinje cells inputs integrity, and more analysis to quantify markers of proliferation, mitosis and apoptosis, confirming no differences between *Tbx1* +/- and WT. They clarified quantification methods, provided more details about the animals used and age groups, and now updated the methods properly for the work to be reproduced.

I only have one really minor suggestion for the authors. In the newly added section: “In contrast, cerebellar (PF/F) cell populations remained stable; no significant shifts (Fig 7h-j, Supplementary Fig. 7h, j) and no change in cell proliferation, cell cycle, or apoptosis was observed between WT and *Tbx1* +/- mice (data not shown)”.

Is the “data not shown” available in a public repository? If so, this could be specified. Or, if referring to the data provided in the rebuttal letter, why not include it in the supplements? Overall, I am satisfied with the revised manuscript and support this work for publication without further revision.

We thank Reviewer 3 for the positive review of the revised manuscript. We have now added the figure that was included in the rebuttal letter as Supplementary Fig. 8, as suggested.

Reviewer #4

I co-reviewed this manuscript with one of the reviewers who provided the listed reports

We thank Reviewer 4 for contributing to the assessment of our paper.

Reviewer #1 (Remarks to the Author):

I thank the authors for their conscientious and complete effort to modify the manuscript based upon my concerns as well as those of the other reviewers.

Reviewer #2 (Remarks to the Author):

In this revised manuscript, the authors adequately addressed all the concerns I had with the previous version. I am in agreement with the authors' view that potential deficits in neural connectivity around Purkinje cells cannot be ruled out to explain the impaired LTD observed in Tbx1^{+/-} or Df(16)1^{+/+} mice. I believe this work is now deemed satisfactory to be accepted for publication.

Reviewer #3 (Remarks to the Author):

authors addressed my previous suggestions improving the manuscript.

Reviewer #4 (Remarks to the Author):

Response to Reviewers

We thank the reviewers for their consideration. We are pleased that our revisions addressed their previous concerns.